# Unified Sample-Optimal Property Estimation in Near-Linear Time

**Yi Hao**
Dept. of Electrical and Computer Engineering
University of California, San Diego
yih179@ucsd.edu

**Alon Orlitsky**
Dept. of Electrical and Computer Engineering
University of California, San Diego
alon@ucsd.edu

## Abstract

We consider the fundamental learning problem of estimating properties of distributions over large domains. Using a novel piecewise-polynomial approximation technique, we derive the first unified methodology for constructing sample- and time-efficient estimators for all sufficiently smooth, symmetric and non-symmetric, additive properties. This technique yields near-linear-time computable estimators whose approximation values are asymptotically optimal and highly-concentrated, resulting in the first: 1) estimators achieving the $\mathcal{O}(k/(\varepsilon^2 \log k))$ min-max $\varepsilon$-error sample complexity for all $k$-symbol Lipschitz properties; 2) unified near-optimal differentially private estimators for a variety of properties; 3) unified estimator achieving optimal bias and near-optimal variance for five important properties; 4) near-optimal sample-complexity estimators for several important symmetric properties over both domain sizes and confidence levels.

## 1 Introduction

Let $\Delta_k$ be the collection of distributions over the alphabet $[k] := \{1, 2, \ldots, k\}$, and $[k]^*$ be the set of finite sequences over $[k]$. In many learning applications, we are given i.i.d. samples $X^n := X_1, X_2, \ldots, X_n$ from an unknown distribution $\vec{p} := (p_1, p_2, \ldots, p_k) \in \Delta_k$, and using these samples we would like to infer certain distribution properties.

A *distribution property* is a mapping $f : \Delta_k \to \mathbb{R}$. Often, these properties are *symmetric* and *additive*, namely, $f(\vec{p}) = \sum_{i \in [k]} f(p_i)$. For example, Shannon entropy, support size, and three more properties in Table 1. Many other important properties are additive but not necessarily symmetric, namely, $f(\vec{p}) = \sum_{i \in [k]} f_i(p_i)$. For example, Kullback-Leibler (KL) divergence or $\ell_1$ distance to a fixed distribution $q$, and distances weighted by the observations $x_i$, e.g., $\sum_{i \in [k]} x_i \cdot |p_i - q_i|$. A *property estimator* is a mapping $\hat{f} : [k]^* \to \mathbb{R}$, where $\hat{f}(X^n)$ approximates $f(\vec{p})$.

Property estimation has attracted significant attention due to its many applications in various disciplines: Shannon entropy is the principal information measure in numerous machine-learning [6] and neurosicence [13] algorithms; support size is essential in population [4] and vocabulary size [29] estimation; support coverage arises in ecological [7], genomic [23], and database [14] studies; $\ell_1$ distance is useful in hypothesis testing [24] and classification [10]; KL divergence reflects the performance of investment [8], compression [9], and on-line learning [20].

For data containing sensitive information, we may need to design special property estimators that preserve individual privacy. Perhaps the most notable notion of privacy is *differential privacy (DP)*. In the context of property estimation [11], we say that an estimator $\hat{f}$ is $\alpha$-*differentially private* if for any $X^n$ and $Y^n$ that differ by at most one symbol, $\Pr(\hat{f}(X^n) \in S)/\Pr(\hat{f}(Y^n) \in S) \le e^{\alpha}$ for any measurable set $S \subset \mathbb{R}$. We consider designing property estimators that achieve small estimation error $\varepsilon$, with probability at least $2/3$, while maintaining $\alpha$-privacy.

Preprint. To appear at NeurIPS 2019.

The next section formalizes our discussion and presents some of the major results in the area.

## 1.1   Problem formulation and prior results

**Property estimation**   Let $f$ be a property over $\Delta_k$. The $(\varepsilon, \delta)$-*sample complexity of an estimator* $\hat{f}$ for $f$ is the smallest number of samples required to estimate $f(\vec{p})$ with accuracy $\varepsilon$ and confidence $1 - \delta$, for all distributions in $\Delta_k$,

$$C_f(\hat{f}, \varepsilon, \delta) := \min\{n : \Pr_{X^n \sim p}(|\hat{f}(X^n) - f(\vec{p})| > \varepsilon) \leq \delta, \ \forall \vec{p} \in \Delta_k\}.$$

The $(\varepsilon, \delta)$-*sample complexity* of estimating $f$ is the lowest $(\varepsilon, \delta)$-sample complexity of any estimator,

$$C_f(\varepsilon, \delta) := \min_{\hat{f}} C_f(\hat{f}, \varepsilon, \delta).$$

Ignoring constant factors and assuming $k$ is large, Table 1 summarizes some of the previous results [2, 22, 26, 31, 33–35] for $\delta = 1/3$. Following the formulation in [2, 31, 34], for support size, we normalize it by $k$ and replace $\Delta_k$ by the collection of distributions $\vec{p} \in \Delta_k$ satisfying $p_i \geq \frac{1}{k}, \forall i \in [k]$. For support coverage [2, 26], the expected number of distinct symbols in $m$ samples, we normalize it by the given parameter $m$ and assume that $m$ is sufficiently large.

Table 1: $C_f(\varepsilon, 1/3)$ for some properties

| Property | $f_i(p_i)$ | $C_f(\varepsilon, 1/3)$ |
|---|---|---|
| Shannon entropy | $p_i \log \frac{1}{p_i}$ | $\frac{k}{\log k} \frac{1}{\varepsilon}$ |
| Power sum of order $a$ | $p_i^a, a < 1$ | $\frac{k^{1/a}}{\varepsilon^{1/a} \log k}$ |
| Distance to uniformity | $\left\| p_i - \frac{1}{k} \right\|$ | $\frac{k}{\log k} \frac{1}{\varepsilon^2}$ |
| Normalized support size | $\frac{\mathbb{1}_{p_i > 0}}{k}$ | $\frac{k}{\log k} \log^2 \frac{1}{\varepsilon}$ |
| Normalized support coverage | $\frac{1 - (1 - p_i)^m}{m}$ | $\frac{m}{\log m} \log \frac{1}{\varepsilon}$ |

**Min-max MSE**   A closely related characterization of an estimator's performance is the *min-max mean squared error (MSE)*. For any unknown distribution $\vec{p} \in \Delta_k$, the MSE of a property estimator $\hat{f}$ in estimating $f(\vec{p})$, using $n$ samples from $\vec{p}$, is

$$R_n(\hat{f}, f, \vec{p}) := \mathbb{E}_{X^n \sim \vec{p}}(\hat{f}(X^n) - f(\vec{p}))^2.$$

Since $\vec{p}$ is unknown, we consider the minimal possible worst-case MSE, or the *min-max MSE*, for any property estimator in estimating property $f$,

$$R_n(f, \Delta_k) := \min_{\hat{f}} \max_{\vec{p} \in \Delta_k} R_n(\hat{f}, f, \vec{p}).$$

The property estimator $\hat{f}^{\mathrm{m}}$ achieving the min-max MSE is the *min-max estimator* [21, 22, 34, 35].

Letting $\vec{p}_{\max} := \arg\max_{\vec{p} \in \Delta_k} R_n(\hat{f}^{\mathrm{m}}, f, \vec{p})$ be the worst-case distribution for $\hat{f}^{\mathrm{m}}$, we can express the min-max MSE as the sum of two quantities: the *min-max squared bias*,

$$\mathrm{Bias}_n^2(\hat{f}^{\mathrm{m}}) := (\mathbb{E}_{X^n \sim \vec{p}_{\max}}[\hat{f}^{\mathrm{m}}(X^n)] - f(\vec{p}_{\max}))^2,$$

and the *min-max variance*,

$$\mathrm{Var}_n(\hat{f}^{\mathrm{m}}) := \mathrm{Var}_{X^n \sim \vec{p}_{\max}}(\hat{f}^{\mathrm{m}}(X^n)).$$

**Private property estimation**   Analogous to the non-private setting above, for an estimator $\hat{f}$ of $f$, let its $(\varepsilon, \delta, \alpha)$-*private sample complexity* $C(\hat{f}, \varepsilon, \delta, \alpha)$ be the smallest number of samples that $\hat{f}$ requires to estimate $f(\vec{p})$ with accuracy $\varepsilon$ and confidence $1 - \delta$, while maintaining $\alpha$-privacy for all distributions $\vec{p} \in \Delta_k$. The $(\varepsilon, \delta, \alpha)$-*private sample complexity* of estimating $f$ is then

$$C_f(\varepsilon, \delta, \alpha) := \min_{\hat{f}} C_f(\hat{f}, \varepsilon, \delta, \alpha).$$

For Shannon entropy, normalized support size, and normalized support coverage, the work of [3] derived tight lower and upper bounds on $C_f(\varepsilon, 1/3, \alpha)$.

## 1.2 Existing methods

There are mainly two types of methods introduced to estimate distribution properties: plug-in, and approximation-empirical, which we briefly discuss below.

**Plug-in**    Major existing plug-in estimators work for only symmetric properties, and in general do not achieve the min-max MSEs' nor the optimal $(\varepsilon, \delta)$-sample complexities. More specifically, the linear-programming based methods initiated by [12], and analyzed and extended in [31–33] achieve the optimal sample complexities only for distance to uniformity and entropy, for relatively large $\varepsilon$. The method basically learns the moments of the underlying distribution from its samples, and finds a distribution whose (low-order) moments are consistent with these estimates. A locally refined version of the linear-programming estimator [15] achieves optimal sample complexities for entropy, power sum, and normalized support size, but requires polynomial time to be computed. This version yields a bias guarantee similar to ours over symmetric properties, yet its variance guarantee is often worse.

Recently, the work of [2] showed that the profile maximum likelihood (PML) estimator [25], an estimator that finds a distribution maximizing the probability of observing the multiset of empirical frequencies, is sample-optimal for estimating entropy, distance to uniformity, and normalized support size and coverage. After the initial submission of the current work, paper [18] showed that the PML approach and its near-linear-time computable variant [5] are sample-optimal for any property that is symmetric, additive, and appropriately Lipschitz, including the four properties just mentioned. This establishes the PML estimator as the first universally sample-optimal plug-in approach for estimating symmetric properties. In comparison, the current work provides a unified property-dependent approach that is sample-optimal for several symmetric and non-symmetric properties.

**Approximation-empirical**    The approximation-empirical method [21, 22, 28, 34, 35] identifies a non-smooth part of the underlying function $f_i$, replaces it by a polynomial $\tilde{f}_i$, and estimates the value of $p_i$ by its empirical frequency $\hat{p}_i$. Depending on whether $\hat{p}_i$ belongs to the non-smooth part or not, the method estimates $f_i(p_i)$ by either the unbiased estimator of $\tilde{f}_i(p_i)$, or the empirical plug-in estimator $f_i(\hat{p}_i)$. However, due to its strong dependency on both the function's structure and the empirical estimator's performance, the method requires significantly different case-by-case modification and analysis, and may not work optimally for general additive properties.

Specifically, 1) The efficacy of this method relies on the accuracy of the empirical plug-in estimator over the smooth segments, which needs to be verified individually for each property; 2) Different functions often have non-smooth segments of different number, locations, and sizes; 3) Combining the non-smooth and smooth segments estimators requires additional care: sometimes needs the knowledge of $k$, sometimes even needs a third estimator to ensure smooth transitioning.

In addition, the method has also not been shown to achieve optimal results for general Lipschitz properties, or many of the other properties covered by the new method in this paper.

## 2 New methodology

The preceding discussion showed that no existing generic method efficiently estimates general additive properties. Motivated by recent advances in the field [2, 15, 16, 19], we derive the first generic method to construct sample-efficient estimators for all sufficiently smooth additive properties.

We start by approximating functions of an unknown Bernoulli success probability from its i.i.d. samples. For a wide class of real functions, we propose a piecewise-polynomial approximation technique, and show that it yields small-bias estimators that are exponentially concentrated around their expected estimates. This provides a different view of property estimation that allows us to simplify the proofs and broaden the range of the results. For details please see Section 4.

**High-level idea**    The idea behind this methodology is natural. By the Chernoff bound for binomial random variables, the empirical count of a symbol in a given sample sequence will not differ from its mean value by too much. Hence, based on the empirical frequency, we can roughly infer which "tiny piece" of $[0, 1]$ the corresponding probability lies in. However, due to randomness, a symbol's empirical frequency may often differ from the true probability value by a small quantity, and plugging it into the function will cause certain amount of bias.

To correct this bias, we first replace the function by its low-degree polynomial approximation *over that "tiny piece"*, and then compute an unbiased estimator of this polynomial. In other words, we use this polynomial as a proxy for the estimation task. We want the degree of the polynomial to be small since this will generally reduce the unbiased estimator's variance; we focus on approximating only a tiny piece of the function because this will reduce the polynomial's approximation error (bias). Given any additive property $f(\vec{p}) = \sum_{i \in [k]} f_i(p_i)$, we apply this technique to each real function $f_i$ and use the corresponding sum to estimate $f(\vec{p})$. Henceforth we use $\hat{f}^*$ to denote this explicit estimator.

## 3    Implications and new results

Because of its conceptual simplicity, the methodology described in the last section has strong implications for estimating all sufficiently smooth additive properties, which we present as theorems.

**Theorem 5** in Section 5 is the root of all the following results, while it is more abstract and illustrating it requires much more effort. Hence for clarity, we begin by presenting several more concrete results.

**Correct asymptotic**    For most of the properties considered in the paper, even the naive empirical-frequency estimator is sample-optimal in the large-sample regime (termed "simple regime" in [34]) where the number of samples $n$ far exceeds the alphabet size $k$. The interesting regime, addressed in numerous recent publications [15, 16, 19, 18, 21, 31, 33, 35], is where $n$ and $k$ are comparable, e.g., differing by at most a logarithmic factor. In this range, $n$ is sufficiently small that sophisticated techniques can help, yet not too small that nothing can be estimated. Since $n$ and $k$ are given, one can decide whether the naive estimator suffices, or sophisticated estimators are needed. For most of the results presented here, the technical significance stems in their nontriviality in this large-alphabet regime. For this reason, we will also assume that $\log k \asymp \log n$ throughout the paper.

### Implication 1: Lipschitz property estimation

An additive property $f(\vec{p}) = \sum_i f_i(p_i)$ is *L-Lipschitz* if all functions $f_i$ have Lipschitz constants uniformly bounded by $L$. Many important properties are Lipschitz, but except for a few isolated examples, it was not known until very recently [16, 19] that general Lipschitz properties can be estimated with sub-linearly many samples. In particular, the result in [16] implies a sample-complexity upper bound of $\mathcal{O}(L^3 k/(\varepsilon^3 \log k))$. We improve this bound to $C_f(\varepsilon, 1/3) \lesssim L^2 k/(\varepsilon^2 \log k)$:

**Theorem 1.** *If $f$ is an L-Lipschitz property, then for any $p \in \Delta_k$ and $X^n \sim p$,*

$$\left| \mathbb{E}\left[ \hat{f}^*(X^n) \right] - f(\vec{p}) \right| \lesssim \sum_{i \in [k]} L\sqrt{\frac{p_i}{n \log n}} \leq L\sqrt{\frac{k}{n \log n}},$$

*and*

$$\mathrm{Var}(\hat{f}^*(X^n)) \leq \mathcal{O}\left( \frac{L^2}{n^{0.99}} \right).$$

This theorem is optimal as even for relatively simple Lipschitz properties, e.g., distance to uniformity (see Table 1 and [22]), the bias bound is optimal up to constant factors, and the variance bound is near-optimal and can not be smaller than $\Omega(L^2/n)$.

### Implication 2: High-confidence property estimation

Surprisingly, the $(\varepsilon, \delta)$-sample complexity has not been fully characterized even for some important properties. A commonly-used approach to constructing an estimator with $(\varepsilon, \delta)$-guarantee is to choose an $(\varepsilon, 1/3)$-estimator, and boost the learning confidence by taking the median of its $\mathcal{O}(\log \frac{1}{\delta})$ independent estimates. This well-known *median trick* yields the following upper bound

$$C_f(\varepsilon, \delta) \lesssim \log \frac{1}{\delta} \cdot C_f(\varepsilon, 1/3).$$

For example, for Shannon entropy,

$$C_f(\varepsilon, \delta) \lesssim \log \frac{1}{\delta} \cdot \frac{k}{\varepsilon \log k} + \log \frac{1}{\delta} \cdot \frac{\log^2 k}{\varepsilon^2}.$$

By contrast, we show that our estimator satisfies

$$C_f(\hat{f}^*, \varepsilon, \delta) \lesssim \frac{k}{\varepsilon \log k} + \left( \log \frac{1}{\delta} \cdot \frac{1}{\varepsilon^2} \right)^{1.01}.$$

To see optimality, Theorem 2 below shows that this upper bound is nearly tight.

In the high-probability regime, namely when $\delta$ is small, the new upper bound obtained using our method could be significantly smaller than the one obtained from the median trick. Theorem 2 shows that this phenomenon also holds for other properties like normalized support size and power sum.

**Theorem 2.** *Ignoring constant factors, Table 2 summarizes relatively tight lower and upper bounds on $C_f(\varepsilon, \delta, k)$ for different properties. In addition, all the upper bounds can be achieved by estimator $\hat{f}^*$.*

Table 2: Bounds on $C_f(\varepsilon, \delta, k)$ for several properties

| Property | Lower bound | Upper bound |
|---|---|---|
| Shannon entropy | $\frac{k}{\varepsilon \log k} + \log \frac{1}{\delta} \cdot \frac{\log^2 k}{\varepsilon^2}$ | $\frac{k}{\varepsilon \log k} + \left(\log \frac{1}{\delta} \cdot \frac{1}{\varepsilon^2}\right)^{1+\beta}$ |
| Power sum of order $a$ | $\frac{k^{\frac{1}{a}}}{\varepsilon^{\frac{1}{a}} \log k} + \log \frac{1}{\delta} \cdot \frac{k^{2-2a}}{\varepsilon^2}$ | $\frac{k^{\frac{1}{a}}}{\varepsilon^{\frac{1}{a}} \log k} + \left[\left(\log \frac{1}{\delta} \cdot \frac{1}{\varepsilon^2}\right)^{\frac{1}{2a-1}}\right]^{1+\beta}$ |
| Normalized support size | $\frac{k}{\log k} \log^2 \frac{1}{\varepsilon}$ | $\frac{k}{\log k} \log^2 \frac{1}{\varepsilon}$ |

*Remarks on Table 2*: Parameter $\beta$ can be any fixed absolute constant in $(0, 1)$. The lower and upper bounds for power sum hold for $a \in (1/2, 1)$. For normalized support size, we require $\delta > \exp(-k^{1/3})$ and $\varepsilon \in (k^{-0.33}, k^{-0.01})$. Note that the restriction on $\varepsilon$ for support-size estimation is imposed only to yield a simple sample-complexity expression. This is not required by our algorithm, which is also sample optimal for $\varepsilon \geq k^{-0.01}$. It is possible that other algorithms can achieve similar upper bounds, while our main point is to demonstrate that our single, unified method has many desired attributes.

## Implication 3: Optimal bias and near-optimal variance

The min-max MSEs of several important properties have been determined up to constant factors, yet there is no explicit and executable scheme for designing estimators achieving them. We show that $\hat{f}^*$ achieves optimal squared bias and near-optimal variance in estimating a variety of properties.

**Theorem 3.** *Up to constant factors, the estimator $\hat{f}^*$ achieves the optimal (min-max) squared bias and near-optimal variance for estimating Shannon entropy, normalized support size, distance to uniformity, and power sum, as well as $\ell_1$ distance to a fixed distribution.*

*Remarks on Theorem 3*: For power sum, we consider the case where the order is less than 1. For normalized support size, we again make the assumption that the minimum nonzero probability of the underlying distribution is at least $1/k$. As noted previously, we consider the parameter regime where $n$ and $k$ are comparable and $k$ is large. In particular, besides the general assumption $\log k \asymp \log n$, we assume that $n \gtrsim k^{1/\alpha} / \log k$ for power sum; $n \gtrsim k / \log k$ for entropy; and $k \log k \gtrsim n \gtrsim k / \log k$ for normalized support size. The proof of the theorem naturally follows from Theorem 5.

## Implication 4: Private property estimation

Privacy is of increasing concern in modern data science. We show that our estimates are exponentially concentrated around the underlying value. Using this attribute we derive a near-optimal differentially-private estimator $\hat{f}^*_{DP}$ for several important properties by adding Laplacian noises to $\hat{f}$.

As an example, for Shannon entropy and properly chosen algorithm hyper-parameters,

$$C_f(\hat{f}^*_{DP}, \varepsilon, 1/3, \alpha) \lesssim \frac{k}{\varepsilon \log k} + \frac{1}{\varepsilon^{2.01}} + \frac{1}{(\alpha\varepsilon)^{1.01}}.$$

This essentially recovers the sample-complexity upper bound in [3], which is nearly tight [3] for all parameters. Hence for large domains, one can achieve strong differential privacy guarantees with only a marginal increase in the sample size compared to the $k/(\varepsilon \log k)$ required for non-private estimation. An analogous argument shows that $\hat{f}^*_{DP}$ is also near-optimal for the private estimation of support size and many others. Section 2.3 of the supplementary material provides more detail as well as unified bounds on the differentially-private sample complexities of general additive properties.

**Outline** The rest of the paper is organized as follows. In Section 4 we construct an estimator that approximates the function value of an unknown Bernoulli success probability, and characterize its performance by Theorem 4. In Section 5 we apply this function estimator to estimating properties of distributions and provide analogous guarantees. Section 6 concludes the paper and also presents possible future directions. We postpone all the proof details to the supplementary material.

## 4 Estimating functions of Bernoulli probabilities

### 4.1 Problem formulation

We begin with a simple problem that involves just a single unknown parameter.

Let $g$ be a continuous real function over the unit interval whose absolute value is uniformly bounded by an absolute constant. Given i.i.d. samples $X^n := X_1, \ldots, X_n$ from a Bernoulli distribution with unknown success probability $p$, we would like to estimate the function value $g(p)$. A *function estimator* is a mapping $\hat{g} : \{0,1\}^* \to \mathbb{R}$. We characterize the performance of the estimator $\hat{g}(X^n)$ in estimating $g(p)$ by its *absolute bias*

$$\text{Bias}(\hat{g}) := |\mathbb{E}[\hat{g}(X^n)] - g(p)|,$$

and *deviation probability*

$$P(\varepsilon) := \Pr(|\hat{g}(X^n) - \mathbb{E}[\hat{g}(X^n)]| > \varepsilon),$$

which implies the variance, and provides additional information useful for property estimation. Our objective is to find an estimator that has near-optimal small bias and Gaussian-type deviation probability $\exp(-n^{\Theta(1)}\varepsilon^2)$ for all possible values of $p \in [0,1]$.

As could be expected, our results are closely related to the smoothness of the function $g$.

### 4.2 Smoothness of real functions

**Effective derivative** Given an interval $I$ and step size $h \in (0, |I|)$, where $|I|$ denotes the interval's length. The *effective derivative* of $g$ over $I$ is the Lipschitz-type ratio

$$L_g(h, I) := \sup_{x,y \in I, |y-x| \geq h} \frac{|g(y) - g(x)|}{|y - x|}.$$

This simple smoothness measure does not fully capture the smoothness of $g$. For example, $g$ could be a zigzag function that has a high effective derivative locally, but over-all fluctuates in only a very small range, and hence is close to a smooth function in maximum deviation. We therefore define a second smoothness measure as the maximum deviation between $g$ and a fixed-degree polynomial. Besides being smooth and having derivatives of all orders, by the Weierstrass approximation theorem, polynomials can also uniformly approximate any continuous $g$.

**Min-max deviation** Let $P_d$ be the collection of polynomials of degree at most $d$. The *min-max deviation* in approximating $g$ over an interval $I$ by a polynomial in $P_d$ is

$$D_g(d, I) := \min_{q \in P_d} \max_{x \in I} |g(x) - q(x)|.$$

The minimizing polynomial is the degree-$d$ min-max polynomial approximation of $g$ over $I$.

For simplicity we abbreviate $L_g(h) := L_g(h, [0,1])$ and $D_g(d) := D_g(d, [0,1])$.

### 4.3 Estimator construction

For simplicity, assume that the sampling parameter is an even number $2n$. Given i.i.d. samples $X^{2n} \sim \text{Bern}(p)$, we let $N_i$ denote the number of times symbol $i \in \{0,1\}$ appears in $X^{2n}$.

We first describe a simplified version of our estimator and provide a non-rigorous analysis relating its performance to the smoothness quantities just defined. The actual more involved estimator and a rigorous performance analysis are presented in Section 1 of the supplementary material.

**High-level description** On a high level, the empirical estimator estimates $g(p)$ by $g(N_1/(2n))$, and often incurs a large bias. To address this, we first partition the unit interval into roughly $\sqrt{n}$ sub-intervals. Then, we split $X^{2n}$ into two halves of equal length $n$ and use the empirical probability of symbol 1 in the first half to identify a sub-interval $I$ and its two neighbors in the partition so that $p$ is contained in one of them, with high confidence. Finally, we replace $g$ by a low-degree min-max polynomial $\tilde{g}$ over $I$ and its four neighbors and estimate $g(p)$ from the second half of the sample sequence by applying a near-unbiased estimator of $\tilde{g}(p)$.

### Step 1: Partitioning the unit interval

Let $\alpha[a, b]$ denote the interval $[\alpha a, \alpha b]$. For an absolute positive constant $c$, define $c_n := c\frac{\log n}{n}$ and a sequence of increasing-length intervals

$$I_j := c_n \left[(j-1)^2, j^2\right], \quad j \geq 1.$$

Observe that the first $M_n := 1/\sqrt{c_n}$ intervals partition the unit interval $[0,1]$. For any $x \geq 0$, we let $j_x$ denote the index $j$ such that $x \in I_j$. This unit-interval partition is directly motivated by the Chernoff bounds. A very similar construction appears in [1], and the exact one appears in [15, 17].

### Step 2: Splitting the sample sequence and locating the probability value

Split the sample sequence $X^{2n}$ into two equal halves, and let $\hat{p}_1$ and $\hat{p}_2$ denote the empirical probabilities of symbol 1 in the first and second half, respectively. By the Chernoff bound of binomial random variables, for sufficiently large $c$, the intervals $I_1, \ldots, I_{M_n}$ form essentially the finest partition of $[0, 1]$ such that if we let $I_j^* := \cup_{j'=j-1}^{j+1} I_{j'}$ and $I_j^{**} := \cup_{j'=j-2}^{j+2} I_{j'}$, then for all underlying $p \notin I_j^*$,

$$\Pr(\hat{p}_1 \in I_j) \leq n^{-3},$$

and for all underlying $p$ and all $j$,

$$\Pr(\hat{p}_1 \in I_j \text{ and } \hat{p}_2 \notin I_j^{**}) \leq n^{-3}.$$

It follows that if $\hat{p}_1 \in I_j$, then with high confidence we can assume that $p \in I_j^*$.

### Step 3: Min-max polynomial estimation

Let $\lambda$ be a universal constant in $(0, 1/4)$ that balances the bias and variance of our estimator. Given the sampling parameter $n$, define

$$d_n := \max\left\{d \in \mathbb{N} : d \cdot 2^{4.5d+2} \leq n^\lambda\right\}.$$

For each $j$, let the *min-max polynomial* of $g$ be the degree-$d_n$ polynomial $\tilde{g}_j$ minimizing the largest absolute deviation with $g$ over $I_j^{**}$.

For each interval $I_j$ we create a piecewise polynomial $\tilde{g}_j^*$ that approximates $g$ over the entire unit interval. It consists of $\tilde{g}_j$ over $I_j^{**}$, and of $\tilde{g}_{j'}$ over $I_{j'}$ for $j' \notin [j-2, j+2]$.

Finally, to estimate $g(p)$, let $j$ be the index such that $\hat{p}_1 \in I_j$, and approximate $\tilde{g}_j^*(p)$ by plugging in unbiased estimators of $p^t$ constructed from $\hat{p}_2$ for all powers $t \leq d_n$. Note that a standard unbiased estimator for $p^t$ is $\prod_{i=0}^{t-1}[(\hat{p}_2 - i/n)/(1 - i/n)]$, and the rest follows from the linearity of expectation.

**Computational complexity** A well-known approximation theory result states that the degree-$d$ truncated Chebyshev series (or polynomial) of a function $g$, often closely approximate the degree-$d$ min-max polynomial of $g$. The Remez algorithm [27, 30] is a popular method for finding such Chebyshev-type approximations of degree $d$, and is often very efficient in practice. Under certain conditions on the function to approximate, running the algorithm for $\log t$ iterations will lead to an error of $\mathcal{O}(\exp(-\Theta(t)))$. Indeed, many state-of-the-art *property* estimators, e.g., [16, 21, 22, 34, 35], use the Remez algorithm to approximate the min-max polynomials, and have implementations that are near-linear-time computable.

## 4.4 Final estimator and its characterization

**The estimator** Consolidating above results, we estimate $g(p)$ by the estimator

$$\hat{g}(\hat{p}_1, \hat{p}_2) := \sum\nolimits_j \hat{g}_j(\hat{p}_2) \cdot \mathbb{1}_{\hat{p}_1 \in I_j}.$$

The exact form and construction of this estimator appear in Section 1.2 of the supplementary material.

**Characterization**  The theorem below characterizes the bias, variance, and mean-deviation probability of the estimator. We sketch its proof here and leave the details to the supplementary material.

According to the reasoning in the last section, for all $p \in I_j^*$, the absolute bias of the resulting estimator $\hat{g}_j(\hat{p}_2)$ in estimating $g(p)$ is essentially upper bounded by $D_g(d_n, I_j^{**})$. Normalizing it by the input's precision $1/n$, we define the (normalized) *local min-max deviation* and the *global min-max deviation* over $I_j^{**}$, respectively, as

$$D_g^*(2n, x) := n \cdot \max_{j' \in [j_x - 1, j_x + 1]} D_g(d_n, I_{j'}^{**}).$$

and

$$D_g^*(2n) := \max_{x' \in [0,1]} D_g^*(2n, x').$$

Hence the bias of $\hat{g}(\hat{p}_1, \hat{p}_2)$ in estimating $g(p)$ is essentially upper bounded by $D_g^*(2n, p)/n \leq D_g^*(2n)/n$. A similar argument yields the following variance bound on $\hat{g}(\hat{p}_1, \hat{p}_2)$, where $D_g^*(2n, p)$ is replaced by the *local effective derivative*,

$$L_g^*(2n, p) := \max_{j' \in [j_p - 1, j_p + 1]} L_g(1/n, I_{j'}^{**}).$$

Analogously, define $L_g^*(2n) := \max_{x \in [0,1]} L_g^*(2n, x)$ as the *global effective derivative*. The mean-deviation probability of this estimator is characterized by

$$S_g^*(2n) := L_g^*(2n) + D_g^*(2n).$$

Specifically, changing one sample in $X^{2n}$ changes the value of $\hat{g}(\hat{p}_1, \hat{p}_2)$ by at most $\Theta(S_g^*(n)n^{\lambda-1})$. Therefore, by McDiarmid's inequality, for any error parameter $\varepsilon$,

$$\Pr(|\hat{g}(\hat{p}_1, \hat{p}_2) - \mathbb{E}[\hat{g}(\hat{p}_1, \hat{p}_2)]| > \varepsilon) \lesssim \exp\left(-\Theta\left(\frac{\varepsilon^2 n^{1-2\lambda}}{S_g^*(2n)^2}\right)\right).$$

**Theorem 4.** *For any bounded function $g$ over $[0, 1]$, $X^n \sim Bern(p)$, and error parameter $\varepsilon > 0$,*

$$|\mathbb{E}[\hat{g}(\hat{p}_1, \hat{p}_2)] - g(p)| \lesssim \frac{p}{n^3} + \frac{D_g^*(n, p)}{n},$$

$$\mathrm{Var}(\hat{g}(\hat{p}_1, \hat{p}_2)) \lesssim \frac{p}{n^5} + \frac{\left(L_g^*(n, p)\right)^2 \cdot p}{n^{1-4\lambda}},$$

*and*

$$\Pr(|\hat{g}(\hat{p}_1, \hat{p}_2) - \mathbb{E}[\hat{g}(\hat{p}_1, \hat{p}_2)]| > \varepsilon) \lesssim \exp\left(-\Theta\left(\frac{\varepsilon^2 n^{1-2\lambda}}{S_g^*(n)^2}\right)\right).$$

Next we use this theorem to derive tight bounds for estimating general additive properties.

## 5   A unified piecewise-polynomial approach to property estimation

Let $f$ be an arbitrary additive property over $\Delta_k$ such that $|f_i(x)|$ is uniformly bounded by some absolute constant for all $i \in [k]$, and $L_\cdot^*(\cdot)$, $D_\cdot^*(\cdot)$, and $S_\cdot^*(\cdot)$ be the smoothness quantities defined in Section 4.3 and 4.4. Let $X^n$ be an i.i.d. sample sequence from an unknown distribution $\vec{p} \in \Delta_k$. Splitting $X^n$ into two sub-sample sequences of equal length, we denote by $\hat{p}_{i,1}$ and $\hat{p}_{i,2}$ the empirical probability of symbol $i \in [k]$ in the first and second sub-sample sequences, respectively.

Applying the technique presented in Section 4, we can estimate the additive property $f(\vec{p}) = \sum_{i \in [k]} f_i(p_i)$ by the estimator $\hat{f}^*(X^n) := \sum_{i \in [k]} \hat{f}_i(\hat{p}_{i,1}, \hat{p}_{i,2})$. Theorem 4 can then be used to show that $\hat{f}^*$ performs well for all sufficiently-smooth additive properties:

**Theorem 5.** *For any $\varepsilon > 0$, $\vec{p} \in \Delta_k$, and $X^n \sim \vec{p}$,*

$$\left|\mathbb{E}\left[\hat{f}^*(X^n)\right] - f(\vec{p})\right| \lesssim \frac{1}{n^3} + \frac{1}{n} \sum_{i \in [k]} D_{f_i}^*(n, p_i),$$

$$\mathrm{Var}(\hat{f}^*(X^n)) \lesssim \frac{1}{n^5} + \frac{1}{n^{1-4\lambda}} \sum_{i \in [k]} \left(L_{f_i}^*(n, p_i)\right)^2 \cdot p_i,$$

*and*

$$\Pr\left(\left|\hat{f}^*(X^n) - \mathbb{E}\left[\hat{f}^*(X^n)\right]\right| > \varepsilon\right) \lesssim \exp\left(-\Theta\left(\frac{\varepsilon^2 n^{1-2\lambda}}{\max_{i \in [k]}(S_{f_i}^*(n))^2}\right)\right).$$

**Discussions** While the significance of the theorem may not be immediately apparent, note that the three equations characterize the estimator's bias, variance, and higher-order moments in terms of the local min-max deviation $D^*_{f_i}(n, p_i)$, the local effective deviation $L^*_{f_i}(n, p_i)$, and the sum of the maximum possible values of the two, $S^*_{f_i}(n)$, respectively. The smoother function $f_i$ is, the smaller $D^*_{f_i}(\cdot)$ and $L^*_{f_i}(\cdot)$ will be. In particular, for simple smooth functions, the values of $D^*$, $L^*$, and $S^*$ can be easily shown to be small, implying that the $f^*$ is nearly optimal under all three criteria.

For example, considering Shannon entropy where $f_i(p_i) = -p_i \log p_i$ for all $i$, we can show that $D^*_{f_i}(n, p_i)$ and $L^*_{f_i}(n, p_i)$ are at most $\mathcal{O}(1/\log n)$ and $\mathcal{O}(\log n)$, respectively. Hence, the bias and variance bounds in Theorem 5 become $k/(n \log n)$ and $(\log n)/n^{1-4\lambda}$, and the tail bound simplifies to $\exp(-\Theta(\varepsilon^2 n^{1-2\lambda}/\log^2 n))$, where $\lambda$ is an arbitrary absolute constant in $(0, 1/4)$, e.g., $\lambda = 0.01$. According to Theorem 2 and results in [21, 35], all these bounds are optimal.

**Computational complexity** We briefly illustrate how our estimator can be computed efficiently in near-linear time in the sample size $n$. As stated in Section 4.3, over each of the $O(\sqrt{n/\log n})$-intervals we constructed, we will find the min-max polynomial of the underlying function over that particular interval, and for many properties, an approximation suffices and the computation takes only poly$(\log n)$ time utilizing the Remez algorithm as previously noted.

Though our construction uses $O(\sqrt{n/\log n})$ such polynomials, for each symbol $i$ *appearing* in the sample sequence $X^n$, we need to compute just one such polynomial to estimate $f_i(p_i)$. The number of symbols appearing in $X^n$ is trivially at most $n$, hence the total time complexity is $O(n \cdot \text{poly}(\log n))$, which is near-linear in $n$. In fact, the computation of all the $O(k\sqrt{n/\log n})$ possible polynomials can be even performed off-line (without samples), and will not affect our estimator's time complexity.

# 6 Conclusion and future directions

We introduced a piecewise min-max polynomial methodology for approximating additive distribution properties. This method yields the first generic approach to constructing sample- and time-efficient estimators for all sufficiently smooth properties. This approach provides the first: 1) sublinear-sample estimators for general Lipschitz properties; 2) general near-optimal private estimators; 3) unified min-max-MSE-achieving estimators for six important properties; 4) near-optimal high-confidence estimators. Unlike previous works, our method covers both symmetric and non-symmetric, differentiable and non-differentiable, properties, under both private and non-private settings.

Two natural extensions are of interest: 1) generalizing the results to properties involving multiple unknown distributions such as distributional divergences; 2) extending the techniques to derive a similarly unified approach for the closely related field of distribution property testing.

Besides the results we established for piecewise polynomial estimators under the min-max estimation framework, the works of [16, 19] recently proposed and studied a different formulation of competitive property estimation that aims to emulate the instance-by-instance performance of the widely used empirical plug-in estimator, using a logarithmic smaller sample size. It is also quite meaningful to investigate the performance of our technique through this new formulation.

### Acknowledgments

We are grateful to the National Science Foundation (NSF) for supporting this work through grants CIF-1564355 and CIF-1619448.

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
