[Supplementary Material · Supplementary.pdf]

# Supplementary Material: Unified Sample-Optimal Property Estimation in Near-Linear Time

**Yi Hao**
Dept. of Electrical and Computer Engineering
University of California, San Diego
yih179@ucsd.edu

**Alon Orlitsky**
Dept. of Electrical and Computer Engineering
University of California, San Diego
alon@ucsd.edu

## Outline

The supplementary material is organized as follows.

Section 1: We address function estimation and proves Theorem 4 in the main paper. Our objective is to design a small-bias estimator whose approximation value is highly concentrated around its mean.

Section 1.1: We present several ancillary results that will be used in subsequent proofs.

Section 1.2: We construct the function estimator $\hat{g}^*$ using piecewise min-max polynomials.

Section 1.3, 1.4, and 1.5: We derive the bias, variance, and tail probability bounds presented in Theorem 4, respectively, showing that the estimator $\hat{g}^*$ admits strong theoretical guarantees for a broad class of functions.

In particular, in Section 1.5.1, we establish a McDiarmid's inequality under Poisson sampling, which is of independent interest.

Section 2: We apply the function estimation technique derived in Section 1 to derive our generic method for learning additive properties, and prove other theorems in the main paper.

Section 2.1: We establish the results in Theorem 5 and show that for all sufficiently smooth properties, our property estimator $\hat{f}^*$ achieves the state-of-the-art performance.

Section 2.2: We consider the problem of estimating Lipschitz properties. By proving Theorem 1, we show for the first time that all Lipschitz properties can be estimated up to a small error $\varepsilon$ using $\mathcal{O}(k/(\varepsilon^2 \log k))$ samples, with probability at least $2/3$.

Section 2.3: We establish a general result on private property estimation, which trivially implies those stated in Section 2.2 of the main paper.

Section 2.4: We utilize Theorem 5 and some specific constructions to prove the upper and lower bounds in Theorem 2, respectively.

# 1 Proof of Theorem 4: Estimating functions of Bernoulli probabilities

## 1.1 Ancillary results

**Useful tools**

The following two lemmas provide tight bounds on the tail probability of a Poisson or binomial random variable. We use these inequalities throughout the proofs.

**Lemma 1** (Chernoff Bound [2]). *Let $X$ be a Poisson or binomial random variable with mean $\mu$, then for any $\delta > 0$,*

$$\mathbb{P}(X \geq (1+\delta)\mu) \leq \left( \frac{e^\delta}{(1+\delta)^{(1+\delta)}} \right)^\mu \leq e^{-(\delta^2 \wedge \delta)\mu/3}$$

*and for any $\delta \in (0,1)$,*

$$\mathbb{P}(X \leq (1-\delta)\mu) \leq \left( \frac{e^{-\delta}}{(1-\delta)^{(1-\delta)}} \right)^\mu \leq e^{-\delta^2 \mu/2}.$$

By setting $\delta$ to be $1/2$ and $1$ in Lemma 1, we have the following corollary.

**Lemma 2.** *Let $X$ be a Poisson or binomial random variable with mean $\mu$, then*

$$\mathbb{P}(X \leq \frac{1}{2}\mu) \leq e^{-0.15\mu}$$

*and*

$$\mathbb{P}(X \geq 2\mu) \leq e^{-0.38\mu}.$$

The $n$-*sensitivity* of an estimator $\hat{f}$ is the maximum possible change in its value when a sample sequence of size-$n$ input sequence is modified at exactly one location,

$$S(\hat{f}, n) := \max\{|\hat{f}(x^n) - \hat{f}(y^n)| : x^n \text{ and } y^n \text{ differ in one location}\}.$$

The McDiarmid's inequality relates $S(\hat{f}, n)$ to the tail probability of $\hat{f}(X^n)$.

**Lemma 3** (McDiarmid's inequality [7]). *Let $\hat{f}$ be an estimator. For any constant $\varepsilon > 0$, distribution $\vec{p} \in \Delta_k$, and i.i.d. sample sequence $X^n \sim \vec{p}$,*

$$\Pr\left( \left| \hat{f}(X^n) - \mathbb{E}[\hat{f}(X^n)] \right| > \varepsilon \right) \leq 2\exp\left( -\frac{2\varepsilon^2}{nS^2(\hat{f}, n)} \right).$$

As illustrated in the main paper, our construction relies on a variety of polynomials. To analyze these polynomials and relate them to other quantities, we often need to bound the polynomials' coefficients based on their ranges. For a real polynomial, the next lemma provides tight upper bounds on the magnitude of its non-constant coefficients.

**Lemma 4.** *Let $p(x) = \sum_{j=0}^d a_j x^j$ be a degree-$d$ real polynomial and*

$$A := \sup_{x_1, x_2 \in [0,1]} |p(x_1) - p(x_2)|,$$

*then for $j \geq 1$,*

$$|a_j| \leq A \cdot 2^{3.5d}.$$

We will utilize the above lemma to bound the variance of polynomial-based estimators.

**Unbiased estimator of $(p - x)^v$ and its characterization**

The following polynomial is related to the unbiased estimator of $(p - x)^v$ under *Poisson sampling*, where we make the sample size an independent Poisson random variable. Note that both $x \in \mathbb{R}$ and $v \in \mathbb{N}$ are given constant parameters.

$$h_{v,x}(y) := \sum_{l=0}^v \binom{v}{l} (-x)^{v-l} \prod_{l'=0}^{l-1} \left( \frac{y}{n} - \frac{l'}{n} \right).$$

This polynomial will play an important role in our consecutive constructions and corresponding proofs. First, we establish and present several useful attributes of $h_{v,x}(y)$ below.

**Lemma 5.** *For a Poisson random variable $Y \sim \mathrm{Poi}(np)$,*
$$\mathbb{E}[h_{v,x}(Y)] = (p-x)^v.$$

*Proof.* By the linearity of expectation and definition of Poisson random variables,

$$
\begin{aligned}
\mathbb{E}[h_{v,x}(Y)] &= \sum_{l=0}^{v} \binom{v}{l} (-x)^{v-l} \mathbb{E}\left[ \prod_{l'=0}^{l-1} \left( \frac{Y}{n} - \frac{l'}{n} \right) \right] \\
&= \sum_{l=0}^{v} \binom{v}{l} (-x)^{v-l} \frac{1}{n^l} \mathbb{E}\left[ \prod_{l'=0}^{l-1} (Y - l') \right] \\
&= \sum_{l=0}^{v} \binom{v}{l} (-x)^{v-l} \frac{e^{-np}}{n^l} \sum_{j=0}^{\infty} \frac{(np)^j}{j!} \prod_{l'=0}^{l-1} (j - l') \\
&= \sum_{l=0}^{v} \binom{v}{l} (-x)^{v-l} \frac{e^{-np}}{n^l} \sum_{j=l}^{\infty} \frac{(np)^j}{(j - l)!} \\
&= \sum_{l=0}^{v} \binom{v}{l} (-x)^{v-l} \frac{(np)^l}{n^l} \left( e^{-np} \sum_{j=l}^{\infty} \frac{(np)^{j-l}}{(j - l)!} \right) \\
&= \sum_{l=0}^{v} \binom{v}{l} (-x)^{v-l} p^l \\
&= (x - p)^v. \qquad \qquad \square
\end{aligned}
$$

Lemma 5 implies that polynomial $h_{v,x}(Y)$ is the unbiased estimator of $(\mathbb{E}[Y]/n - x)^v$ for $Y \sim \mathrm{Poi}(\cdot)$. The next three lemmas bound the polynomial's value when the input variable is close to its expectation.

**Lemma 6.** *For a Poisson random variable $Y \sim \mathrm{Poi}(np)$,*
$$\mathbb{E}[h_{v,0}^2(Y)] \le \frac{\mathbb{E}[Y^{2v}]}{n^{2v}}.$$
*Furthermore, if for some positive constant $c'$, both $np$ and $2v$ are at most $\le c' \log n$,*
$$\mathbb{E}[h_{v,0}^2(Y)] \le 2p \left( \frac{2c' \log n}{n} \right)^{2v-1}.$$

*Proof.* We consider the first inequality. Note that for all $y \in \mathbb{Z}^+$,
$$0 \le \prod_{l'=0}^{v-1} (y - l') = \mathbb{1}_{y \ge v} \cdot \prod_{l'=0}^{v-1} (y - l') \le y^v.$$

This inequality trivially implies that

$$\mathbb{E}[h_{v,0}^2(Y)] = \frac{1}{n^{2v}} \mathbb{E}\left( \prod_{l'=0}^{v-1} (Y - l') \right)^2 \le \frac{\mathbb{E}[Y^{2v}]}{n^{2v}}.$$

Based on the first inequality, we prove the second one as follows.

$$
\begin{aligned}
\mathbb{E}[h_{v,0}^2(Y)] &\le \frac{\mathbb{E}[Y^{2v}]}{n^{2v}} \\
&\le \frac{1}{n^{2v}} \sum_{t=1}^{2v} t^{2v-t} \binom{2v}{t} (np)^t \\
&\le \frac{1}{n^{2v}} \sum_{t=1}^{2v} (2v)^{2v-t} \binom{2v}{t} (c' \log n)^t \frac{np}{c' \log n} \\
&\le \frac{1}{n^{2v}} (2v + c' \log n)^{2v} \frac{np}{c' \log n} \\
&\le 2p \left( \frac{2c' \log n}{n} \right)^{2v-1}. \qquad \qquad \square
\end{aligned}
$$

**Lemma 7.** *[6] For a Poisson random variable $Y \sim \mathrm{Poi}(np)$ and a parameter*

$$M \geq \max\left\{\frac{n(p-x)^2}{p}, v\right\},$$

*we have*

$$\mathbb{E}[h_{v,0}^2(Y)] \leq \left(\frac{2Mp}{n}\right)^v.$$

**Lemma 8.** *[4] For $x \in [0,1]$, $v \in \mathbb{N}$, $m \in \mathbb{N}$, and a parameter*

$$\delta \geq \max\left\{\left|x - \frac{m}{n}\right|, \frac{\sqrt{4mv}}{n}\right\},$$

*we have*

$$|h_{v,x}(m)| \leq (2\delta)^v.$$

## 1.2 Function estimator construction

Let $g$ be a continuous real function over the unit interval. Given i.i.d. samples $X^n$ from a Bernoulli distribution with unknown success probability $p$, our objective is to estimate the function value $g(p)$.

**Poisson sampling and sample splitting** Generating exactly $n$ samples creates dependencies between the counts of symbols. To simplify the derivations, we use the well-known *Poisson sampling* technique and make the sample size an independent Poisson variable $N$ with mean $n$. In addition, we apply the standard *sample splitting* method and divide the sample sequence $X^N$ into two sub-sample sequences by independently putting each sample into one of the two with equal probability. Equivalently, we can simply generate two independent sample sequences from $\mathrm{Bern}(p)$, each of an independent $\mathrm{Poi}(n/2)$ size. For notational convenience, we replace $n$ by $2n$ and denote by $N_1$ and $N_1'$ the number of times symbol 1 appearing in the first and second sample sequences, respectively.

**Covering the unit interval** Let $c$ be a sufficiently large constant and define $c_n := c\frac{\log n}{n}$. Cover the unit interval $[0,1]$ by three sets of nested intervals

$$I_j := c_n \left[(j-1)^2, j^2\right],$$

$$I_j^* := c_n \left[(j-2)^2 \mathbb{1}_{j\geq 2}, (j+1)^2\right] = \bigcup_{j'=j-1}^{j+1} I_{j'},$$

$$I_j^{**} := c_n \left[(j-3)^2 \mathbb{1}_{j\geq 3}, (j+2)^2\right] = \bigcup_{j'=j-1}^{j+1} I_{j'}^*,$$

where $j = 1, \ldots$ and in the union, $I_{-2}$ and $I_{-1}$ are taken to be empty.

Let $M_n := 1/\sqrt{c_n}$ be the number of intervals so that $I_1, \ldots, I_{M_n}$ form a partition of $[0,1]$.

Parameter $c$ and these intervals are chosen so that for all $j \in [M_n]$, if $N_1/n \in I_j$ we can assume that $p \in I_j^*$ and $N_1'/n \in I_j^{**}$, and regardless of the value of $p$, with high probability we will be right.

**Min-max polynomial approximation** For each $j \in [M_n]$, let $x_j := c_n(j-3)^2 \mathbb{1}_{j\geq 3}$ be the left end point of $I_j^{**}$, and $|I_j^{**}| := c_n(j+2)^2 - c_n(j-3)^2 \mathbb{1}_{j\geq 3}$ be the length of the interval $I_j^{**}$.

Then for any $x \in I_j^{**}$, there exists $y_x \in [0,1]$ such that $x = x_j + |I_j^{**}| \cdot y_x$. Let $\lambda$ be a small absolute constant in $(0, 0.1)$, and define the *degree parameter* as

$$d_n := \max\left\{d \in \mathbb{N}: d \cdot 2^{4.5d+2} \leq n^\lambda\right\}.$$

Denoting

$$r_j(y) := g\left(x_j + |I_j^{**}|y\right),$$

we can find the degree-$d_n$ min-max polynomial of $r_j(y)$ over $y \in [0,1]$, say

$$\tilde{r}_j(y) := \sum_{v=0}^{d_n} a_{jv} y^v.$$

By Lemma 4, for all $v \geq 1$, the following upper bound on $|a_{jv}|$ holds.

$$|a_{jv}| \leq 2^{3.5 d_n} \sup_{z_1, z_2 \in I_j^{**}} |g(z_1) - g(z_2)|.$$

Noting that $y_x = |I_j^{**}|^{-1}(x - x_j)$, we can re-write $\tilde{r}_j(y_x)$ as

$$\tilde{g}_j(x) := \sum_{v=0}^{d_n} a_{jv} |I_j^{**}|^{-v}(x - x_j)^v.$$

**Piecewise-polynomial estimator $\hat{g}^*$**  By Lemma 5, for $j \in [M_n]$, an unbiased estimator of $\tilde{g}_j(p)$ is

$$E_{\tilde{g}_j}(N_1) := \sum_{v=0}^{d_n} a_{jv} |I_j^{**}|^{-v} h_{v,x_j}(N_1) = \sum_{v=0}^{d_n} a_{jv} |I_j^{**}|^{-v} \sum_{l=0}^{v} \binom{v}{l} (-x_j)^{v-l} \prod_{l'=0}^{l-1} \left( \frac{N_1}{n} - \frac{l'}{n} \right).$$

For $j > M_n$, we denote

$$E_{\tilde{g}_j}(N_1) := E_{\tilde{g}_{M_n}} \left( \min \left\{ N_1, c_n(M_n + 2)^2 \right\} \right).$$

Let $T$ be a sufficiently large constant satisfying $T \gg \max_{x \in [0,1]} |g(x)|$, and write $[A]_a^b$ instead of $\min\{\max\{A, a\}, b\}$. Utilizing sample splitting, we estimate $g(p)$ by the following estimator,

$$\hat{g}^*(N_1, N_1') := \left[ \sum_{j=1}^{\infty} \left( E_{\tilde{g}_j}(N_1) \mathbb{1}_{\frac{N_1}{n} \in I_j^{**}} + \sum_{j' \notin [j-2:j+2]} E_{\tilde{g}_{j'}}(N_1) \mathbb{1}_{\frac{N_1}{n} \in I_{j'}} \right) \mathbb{1}_{\frac{N_1'}{n} \in I_j} \right]_{-T}^{T}.$$

## 1.3  Bounding the bias of $\hat{g}^*$

Recall that $I_1, \ldots, I_{M_n}$ form a partition of $[0,1]$. For any $x \in [0,1]$, let $j_x$ denote the index $j$ such that $x \in I_j$. By the triangle inequality, the absolute bias of $\hat{g}^*(N_1, N_1')$ admits

$$|\mathbb{E}[\hat{g}^*(N_1, N_1')] - g(p)| \leq \left| \tilde{g}_{j_p}(p) - g(p) \right| + \left| \mathbb{E} \left[ \hat{g}^*(N_1, N_1') - \tilde{g}_{j_p}(p) \right] \right|$$

$$\leq \left| \tilde{g}_{j_p}(p) - g(p) \right| + \mathbb{E} \left[ 2T \left( \mathbb{1}_{\frac{N_1}{n} \notin I_{j_p}^*} \mathbb{1}_{\frac{N_1'}{n} \in I_{j_p}^*} + \mathbb{1}_{\frac{N_1'}{n} \notin I_{j_p}^*} \right) \right]$$

$$+ \left| \mathbb{E} \left[ \left( \hat{g}^*(N_1, N_1') - \tilde{g}_{j_p}(p) \right) \mathbb{1}_{\frac{N_1}{n} \in I_{j_p}^*} \mathbb{1}_{\frac{N_1'}{n} \in I_{j_p}^*} \right] \right|.$$

The last summation has three terms. By definition, the first term is no larger than $D_g^*(n,p)/n$. By the Chernoff bound (Lemma 1) and the fact that $p \in I_{j_p}$, for sufficiently large constant $c$, the second term is at most $Tp/n^5$. Therefore, it remains to consider the third term. By the triangle inequality and definition of $\hat{g}^*$, the third term is at most

$$B_n(g, p) := \left| \mathbb{E} \left[ \left( \tilde{g}_{j_p - 1}(p) - \tilde{g}_{j_p}(p) \right) \mathbb{1}_{\frac{N_1}{n} \in I_{j_p}^*} \right] \right| + \left| \mathbb{E} \left[ \left( \tilde{g}_{j_p + 1}(p) - \tilde{g}_{j_p}(p) \right) \mathbb{1}_{\frac{N_1}{n} \in I_{j_p}^*} \right] \right|$$

$$+ \left| \mathbb{E} \left[ \left( E_{\tilde{g}_{j_p}}(N_1) - \tilde{g}_{j_p}(p) \right) \mathbb{1}_{\frac{N_1}{n} \in I_{j_p}^*} \right] \right| + \left| \mathbb{E} \left[ \left( E_{\tilde{g}_{j_p - 1}}(N_1) - \tilde{g}_{j_p - 1}(p) \right) \mathbb{1}_{\frac{N_1}{n} \in I_{j_p}^*} \right] \right|$$

$$+ \left| \mathbb{E} \left[ \left( E_{\tilde{g}_{j_p + 1}}(N_1) - \tilde{g}_{j_p + 1}(p) \right) \mathbb{1}_{\frac{N_1}{n} \in I_{j_p}^*} \right] \right|.$$

We bound the first term of $B_n(g, p)$ as

$$\left| \mathbb{E} \left[ \left( \tilde{g}_{j_p - 1}(p) - \tilde{g}_{j_p}(p) \right) \mathbb{1}_{\frac{N_1}{n} \in I_{j_p}^*} \right] \right| \leq \left| \mathbb{E} \left[ \tilde{g}_{j_p - 1}(p) - \tilde{g}_{j_p}(p) \right] \right|$$

$$\leq \left| \tilde{g}_{j_p - 1}(p) - g(p) \right| + \left| g(p) - \tilde{g}_{j_p}(p) \right|$$

$$\leq \frac{2 D_g^*(2n, p)}{n},$$

where the last step follows from the definition of $D_g^*(2n, p)$. The second term of $B_n(g, p)$ satisfies the same inequality, and is at most $2D_g^*(2n, p)/n$. Note that the last three terms of $B_n(g, p)$ are clearly of the same type. Hence for simplicity, below we only analyze the first one.

For any $j \in [M_n]$, we can express $E_{\tilde{g}_j}(N_1)$ in terms of $h_{v,x_j}(N_1)$, i.e.,

$$E_{\tilde{g}_j}(N_1) = \sum_{v=0}^{d_n} a_{jv} |I_j^{**}|^{-v} h_{v,x_j}(N_1).$$

In addition, recall that by definition,

$$\tilde{g}_j(p) = \sum_{v=0}^{d_n} a_{jv} |I_j^{**}|^{-v} (p - x_j)^v.$$

The linearity of expectation combines the above two equalities and yields

$$\mathbb{E}\left[\left(E_{\tilde{f}_j}(N_1) - \tilde{f}_j(p)\right) \mathbb{1}_{\frac{N_1}{n} \in I_j^*}\right] = \sum_{v=0}^{d_n} a_{jv} |I_j^{**}|^{-v} \mathbb{E}\left[\left(h_{v,x_j}(N_1) - (p - x_j)^v\right) \mathbb{1}_{\frac{N_1}{n} \in I_j^*}\right].$$

Therefore, given integers $a$ and $b$ satisfying $b > a > d_n$, our *new objective* is to bound

$$IN_{v,n}(a, b, p, j) := \mathbb{E}\left[\left(h_{v,x_j}(N_1) - (p - x_j)^v\right) \mathbb{1}_{N_1 \in [a,b]}\right].$$

**Bounding the magnitude of $IN_{v,n}$**    For all integer $s \geq 1$, let us denote

$$H_{v,n}(s, p, j) := \sum_{l=0}^{v} \binom{v}{l} (-x_j)^{v-l} p^l \sum_{t \in [s-l, s-1]} e^{-np} \frac{(np)^t}{t!}.$$

We first relate $IN_{v,n}$ to $H_{v,n}$ through the following lemma.

**Lemma 9.** *For any two integers $a$ and $b$ satisfying $a > b > v$,*

$$IN_{v,n}(a, b, p, j) = H_{v,n}(a, p, j) - H_{v,n}(b + 1, p, j).$$

*Proof.* By the linearity of expectation and binomial theorem, we can rewrite the left-hand side as

$$IN_{v,n}(a, b, p, j) = \sum_{l=0}^{v} \binom{v}{l} (-x_j)^{v-l} \mathbb{E}\left[\left(\prod_{l'=0}^{l-1}\left(\frac{N_1}{n} - \frac{l'}{n}\right) - p^l\right) \mathbb{1}_{N_1 \in [a,b]}\right].$$

For each $l \leq v$, we evaluate the inner expectation as follows:

$$\mathbb{E}\left[\left(\prod_{l'=0}^{l-1}\left(\frac{N_1}{n} - \frac{l'}{n}\right) - p^l\right) \mathbb{1}_{N_1 \in [a,b]}\right] = \sum_{t \in [a,b]} \prod_{l'=0}^{l-1}\left(\frac{t}{n} - \frac{l'}{n}\right) e^{-np} \frac{(np)^t}{t!} - p^l \sum_{t \in [a,b]} e^{-np} \frac{(np)^t}{t!}$$

$$= \sum_{t \in [a,b]} \frac{1}{n^l} \frac{t!}{(t-l)!} e^{-np} \frac{(np)^t}{t!} - p_i^l \sum_{t \in [a,b]} e^{-np} \frac{(np)^t}{t!}$$

$$= p^l \sum_{t \in [a-l, b-l]} e^{-np} \frac{(np)^t}{t!} - p^l \sum_{t \in [a,b]} e^{-np} \frac{(np)^t}{t!}$$

$$= p^l \sum_{t \in [a-l, a-1]} e^{-np} \frac{(np)^t}{t!} - p^l \sum_{t \in [b-l+1, b]} e^{-np} \frac{(np)^t}{t!}. \qquad \square$$

Therefore, to bound $|IN_{v,n}(a, b, p, j)|$, we only need to bound $|H_{v,n}(a, p, j)|$ and $|H_{v,n}(b + 1, p, j)|$. We shall proceed by relating these quantities to $h_{l,x_j}(a - 1)$ for $l = 0, \ldots, v - 1$.

**Lemma 10.** *For any integer $s$ satisfying $s > v$,*

$$H_{v,n}(s, p, j) = p e^{-np} \frac{(np)^{s-1}}{(s-1)!} \sum_{l=0}^{v-1} (p - x_j)^{v-l-1} h_{l,x_j}(s - 1).$$

*Proof.* The following recursive formula of binomial coefficients is well-known:

$$\binom{v}{l} = \binom{v-1}{l} + \binom{v-1}{l-1},$$

Utilizing this recursive formula, we can re-write the quantity of interest as

$$H_{v,n}(s,p,j) = \sum_{l=0}^{v-1} \binom{v-1}{l}(-x_j)^{v-(l+1)}p^{l+1} \sum_{t\in[s-l,s-1]} e^{-np}\frac{(np)^t}{t!}$$

$$+ \sum_{l=0}^{v-1} \binom{v-1}{l}(-x_j)^{v-l}p^l \sum_{t\in[s-l,s-1]} e^{-np}\frac{(np)^t}{t!}$$

$$+ \sum_{l=0}^{v-1} \binom{v-1}{l}(-x_j)^{v-(l+1)}p^{l+1}e^{-np}\frac{(np)^t}{t!}\Bigg|_{t=s-(l+1)}$$

$$= (p-x_j)\sum_{l=0}^{v-1} \binom{v-1}{l}(-x_j)^{(v-1)-l}p^l \sum_{t\in[s-l,s-1]} e^{-np}\frac{(np)^t}{t!}$$

$$+ \sum_{l=0}^{v-1} \binom{v-1}{l}(-x_j)^{v-(l+1)}p^{l+1}e^{-np}\frac{(np)^{s-(l+1)}}{(s-(l+1))!}$$

$$= (p-x_j)H_{v-1,n}(s,p,j) + \sum_{l=0}^{v-1} \binom{v-1}{l}(-x_j)^{v-(l+1)}p^{l+1}e^{-np}\frac{(np)^{s-(l+1)}}{(s-(l+1))!}.$$

This equation establishes a standard recursive relation between $H_{v,n}(s,p,j)$ and $H_{v-1,n}(s,p,j)$. To prove our desired result, we relate the second quantity on the right-hand side to $h_{v-1,x_j}(s-1)$,

$$\sum_{l=0}^{v-1} \binom{v-1}{l}(-x_j)^{v-(l+1)}p^{l+1}e^{-np}\frac{(np)^{s-(l+1)}}{(s-(l+1))!}$$

$$= e^{-np}p\sum_{l=0}^{v-1} \binom{v-1}{l}(-x_j)^{(v-1)-l}p_x^l\frac{(np)^{(s-1)-l}}{((s-1)-l)!}$$

$$= e^{-np}p\sum_{l=0}^{v-1} \binom{v-1}{l}(-x_j)^{(v-1)-l}\frac{1}{n^l}\frac{(np)^{s-1}}{((s-1)-l)!}$$

$$= e^{-np}p\sum_{l=0}^{v-1} \binom{v-1}{l}(-x_j)^{(v-1)-l}\frac{\prod_{l'=0}^{l-1}((s-1)-l')}{n^l}\frac{(np)^{s-1}}{(s-1)!}$$

$$= e^{-np}p\frac{(np)^{s-1}}{(s-1)!}\sum_{l=0}^{v-1} \binom{v-1}{l}(-x_j)^{(v-1)-l}\prod_{l'=0}^{l-1}\left(\frac{s-1}{n}-\frac{l'}{n}\right)$$

$$= pe^{-np}\frac{(np)^{s-1}}{(s-1)!}h_{v-1,x_j}(s-1).$$

Substituting the last quantity into the previous recursive relation yields

$$H_{v,n}(s,p,j) = (p-x_j)H_{v-1,n}(s,p,j) + pe^{-np}\frac{(np)^{s-1}}{(s-1)!}h_{v-1,x_j}(s-1),$$

with a base case $H_{0,n}(s,p,j) = 0$. Therefore, the principle of mathematical induction implies

$$H_{v,n}(s,p,j) = pe^{-np}\frac{(np)^{s-1}}{(s-1)!}\sum_{l=0}^{v-1}(p-x_j)^{v-l-1}h_{l,x_j}(s-1). \qquad \square$$

Without loss of generality, we assume that $c\log n$ is a positive integer so that $nx_j \in \mathbb{Z}^+$ for all $j$, since otherwise we can modify the value of $c$ by at most 1 to fulfill this assumption. As an implication of Lemma 10, for integer $s$ such that $s/n$ or $(s-1)/n$ is the end point of $I_{j_p}^*$ (right end point if

$j_p \leq 2$), and sufficiently large constant $c$ satisfying $c \log n > d$,

$$
\begin{aligned}
|H_{v,n}(s,p,j_p)| &= \left| p e^{-np} \frac{(np)^{s-1}}{(s-1)!} \sum_{l=0}^{v-1} (p - x_{j_p})^{v-l-1} h_{l,x_{j_p}}(s-1) \right| \\
&= \Pr(N_1 = s - 1) \cdot p \left| \sum_{l=0}^{v-1} (p - x_{j_p})^{v-l-1} h_{l,x_{j_p}}(s-1) \right| \\
&\leq \frac{p}{n^5} \left| \sum_{l=0}^{v-1} (p - x_{j_p})^{v-l-1} h_{l,x_{j_p}}(s-1) \right| \\
&\leq \frac{p}{n^5} v \left( 2|I_{j_p}^{**}| \right)^{v-1},
\end{aligned}
$$

where the second last step follows from the Chernoff bound and the last step follows from Lemma 8 by setting $\delta = |I_{j_p}^{**}|$. Under the same set of conditions, we can show that

$$
|H_{v,n}(s,p,j_p - 1)| \leq \frac{p}{n^5} v \left( 2|I_{j_p}^{**}| \right)^{v-1}
$$

and

$$
|H_{v,n}(s,p,j_p + 1)| \leq \frac{p}{n^5} v \left( 2|I_{j_p}^{**}| \right)^{v-1}.
$$

**Bounding the bias of $\hat{g}^*$**    Now we are ready to analyze the quantity of interest:

$$
\begin{aligned}
\left| \mathbb{E}\left[ \left( E_{\tilde{g}_{j_p}}(N_1) - \tilde{g}_{j_p}(p) \right) \mathbb{1}_{\frac{N_1}{n} \in I_{j_p}^*} \right] \right| &= \left| \sum_{v=0}^{d_n} a_{j_p v} |I_{j_p}^{**}|^{-v} \mathbb{E}\left[ \left( h_{v,x_{j_p}}(N_1) - (p - x_{j_p})^v \right) \mathbb{1}_{\frac{N_1}{n} \in I_{j_p}^*} \right] \right| \\
&= \left| \sum_{v=0}^{d_n} a_{j_p v} |I_{j_p}^{**}|^{-v} I N_{v,n}(n x_{j_p+1}, n x_{j_p+4}, p, j_p) \right| \\
&= \left| \sum_{v=0}^{d_n} a_{j_p v} |I_{j_p}^{**}|^{-v} (H_{v,n}(n x_{j_p+1}, p, j) \mathbb{1}_{j_p > 2} - H_{v,n}(n x_{j_p+4} + 1, p, j)) \right| \\
&\leq \sum_{v=1}^{d_n} a_{j_p v} |I_{j_p}^{**}|^{-v} \frac{2p}{n^5} v \left( 2|I_{j_p}^{**}| \right)^{v-1} \\
&\leq \sum_{v=1}^{d_n} 2T \cdot 2^{3.5 d_n} \left( \frac{1}{4 c_n} \right) \frac{2p}{n^5} v (2)^{v-1} \\
&\leq \frac{T d_n \cdot 2^{4.5 d_n}}{c_n n^5} \cdot p.
\end{aligned}
$$

The same reasoning also shows that

$$
\left| \mathbb{E}\left[ \left( E_{\tilde{g}_{j_p-1}}(N_1) - \tilde{g}_{j_p-1}(p) \right) \mathbb{1}_{\frac{N_1}{n} \in I_{j_p}^*} \right] \right| \leq \frac{T d_n \cdot 2^{4.5 d_n}}{c_n n^5} \cdot p
$$

and

$$
\left| \mathbb{E}\left[ \left( E_{\tilde{g}_{j_p+1}}(N_1) - \tilde{g}_{j_p+1}(p) \right) \mathbb{1}_{\frac{N_1}{n} \in I_{j_p}^*} \right] \right| \leq \frac{T d_n \cdot 2^{4.5 d_n}}{c_n n^5} \cdot p.
$$

Consolidating all the previous results yields the desired bias bound:

$$
\begin{aligned}
|\mathbb{E}[\hat{g}^*(N_1, N_1')] - g(p)| &\leq \frac{T}{n^5} \cdot p + \frac{3 T d_n \cdot 2^{4.5 d_n}}{c_n n^5} \cdot p + \frac{5}{n} D_g^*(2n, p) \\
&\leq \frac{p}{n^{5-\lambda}} + \frac{5}{n} D_g^*(2n, p).
\end{aligned}
$$

## 1.4 Bounding the variance of $\hat{g}^*$

In this section, we establish the following bound on the variance of our estimator.

**Lemma 11.** *For sufficiently large $c$,*

$$\mathrm{Var}(\hat{g}^*(N_1, N_1')) \leq \frac{72c(\log n)}{n^{1-3\lambda}} \left(L_g^*(2n, p)\right)^2 \cdot p + \frac{8T^2}{n^5} \cdot p.$$

*Proof.* Since $\mathrm{Var}(X) \leq \mathbb{E}[X^2]$ and $\mathbb{1}_X \cdot \mathbb{1}_{\overline{X}} = 0$ for any random variable $X$, we have

$$\mathrm{Var}(\hat{g}^*(N_1, N_1')) \leq \mathbb{E}\left(\hat{g}^*(N_1, N_1')\mathbb{1}_{\frac{N_1}{n} \in I_{j_p}^*}\mathbb{1}_{\frac{N_1'}{n} \in I_{j_p}^*}\right)^2$$

$$+ \mathbb{E}\left(\hat{g}^*(N_1, N_1')\left(1 - \mathbb{1}_{\frac{N_1}{n} \in I_{j_p}^*}\mathbb{1}_{\frac{N_1'}{n} \in I_{j_p}^*}\right)\right)^2$$

$$\leq \sum_{j' \in [j_p - 1, j_p + 1]} \mathbb{E}\left(\hat{g}^*(N_1, N_1')\mathbb{1}_{\frac{N_1}{n} \in I_{j_p}^*}\mathbb{1}_{\frac{N_1'}{n} \in I_{j'}}\right)^2$$

$$+ 4T^2 \cdot \Pr\left(\frac{N_1}{n} \notin I_{j_p}^* \text{ or } \frac{N_1'}{n} \notin I_{j_p}^*\right).$$

$$\leq \sum_{j' \in [j_p - 1, j_p + 1]} \mathbb{E}[E_{\hat{g}_{j'}}^2(N_1)] + 8T^2 \cdot \Pr\left(\frac{N_1}{n} \notin I_{j_p}^*\right).$$

For sufficiently large $c$, the second term is at most $8T^2 p/n^5$ by the Chernoff bound. It remains to analyze $\mathbb{E}[E_{\hat{g}_j}^2(N_1)]$ for $j \in [j_p - 1, j_p + 1]$. By the Cauchy-Schwarz inequality,

$$\mathbb{E}[E_{\hat{g}_j}^2(N_1)] = \mathbb{E}\left(\sum_{v=0}^{d_n} a_{jv}|I_j^{**}|^{-v} h_{v,x_j}(N_1)\right)^2$$

$$\leq \left(\sum_{v=0}^{d_n} |a_{jv}||I_j^{**}|^{-v}\left(\mathbb{E}[h_{v,x_j}^2(N_1)]\right)^{\frac{1}{2}}\right)^2.$$

Consider the inner expectation. If $j_{p_i} \leq 2$ and $j \in [j_{p_i} - 1, j_{p_i} + 1]$, then $x_j = 0$. By Lemma 6,

$$\mathbb{E}[h_{v,x_j}^2(N_1)] \leq \frac{2(32c\log n)^{2v-1}p}{n^{2v-1}}.$$

This together with Lemma 4 and the definition of $L_g^*(n, p)$ implies that

$$\mathbb{E}[E_{\hat{g}_j}^2(N_1)] \leq \left(\sum_{v=0}^{d_n} |a_{jv}||I_j^{**}|^{-v}\left(\mathbb{E}[h_{v,x_j}^2(N_1)]\right)^{\frac{1}{2}}\right)^2$$

$$\leq \left(\sum_{v=0}^{d_n} \left(2^{3.5d_n+1} L_g^*(2n, p)|I_j^{**}|\right)|I_j^{**}|^{-v}\left(\frac{32c\log n}{n}\right)^{v-1}\sqrt{\frac{64c(\log n)p}{n}}\right)^2$$

$$\leq \left(d_n 2^{5.5d_n+1} L_g^*(2n, p)\right)^2 \frac{64c(\log n)p}{n}.$$

If $j_p > 2$ and $j \in [j_p - 1, j_p + 1]$, then by Lemma 7,

$$\mathbb{E}[h_{v,x_j}^2(N_1)] \leq \left(\frac{72c(\log n)p}{n}\right)^v \leq \left(\frac{72c^2(\log n)^2 j_p^2}{n^2}\right)^{v-1}\left(\frac{72c(\log n)p}{n}\right).$$

Analogously,

$$\mathbb{E}[E_{\hat{g}_j}^2(N_1)] \leq \left(\sum_{v=0}^{d_n} |a_{jv}||I_j^{**}|^{-v}\left(\mathbb{E}[h_{v,x_j}^2(N_1)]\right)^{\frac{1}{2}}\right)^2$$

$$\leq \left(\sum_{v=0}^{d_n} \left(2^{3.5d_n+1} L_g^*(2n, p)|I_j^{**}|\right)|I_j^{**}|^{-v}\left(\frac{3 \cdot 2^{1.5}c(\log n)j_p}{n}\right)^{v-1}\sqrt{\frac{72c(\log n)p}{n}}\right)^2$$

$$\leq \left(d_n 2^{4.5d_n+1} L_g^*(2n, p)\right)^2 \frac{72c(\log n)p}{n}.$$

Consolidating the above results yields the desired bound. $\qquad\square$

## 1.5 Sensitivity bound

Incorporate our sampling scheme, we define the *sensitivity* of an estimator $\hat{g}$ as the maximum possible change in its value when an input sequence is replaced by another that differs in exactly one location,

$$S(\hat{g}) := \max\left\{\left|\hat{g}(x^m) - \hat{g}(y^{m'})\right| : m, m' \in \mathbb{Z}, \ x^m \text{ and } y^{m'} \text{ differ in one location}\right\}.$$

By construction, sensitivity upperly bounds $n$-sensitivity, i.e., $S(\hat{g}) \geq S(\hat{g}, n)$ for all $n$. Due to sample splitting, replacing the given sample sequence $X^N$ by a sequence that differs in at most one location could change $N_1$, $N_1'$, or both, by at most one. In other words, to bound the sensitivity of $\hat{g}^*$, we need to bound the change in the estimator's value when we modify $N_1$ or $N_1'$ by one. We proceed as follows. If the value of $N_1$ increases or decreases by one, we need to consider the following two types of differences:

$$\mathbb{D}_g^{(1)}(n, j, s) := E_{\tilde{g}_j}(s) - E_{\tilde{g}_j}(s-1),$$

for $s$ satisfying $s - 1, s,$ or $s + 1 \in nI_j^{**}$, and

$$\mathbb{D}_g^{(2)}(n, j, s) := E_{\tilde{g}_j}(s) - E_{\tilde{g}_{j-1}}(s-1),$$

for $s$ satisfying $s \in nI_{j-1}^{**} \cap nI_j^{**}$. If the value of $N_1'$ increases or decreases by one, we need to consider the difference:

$$\mathbb{D}_g^{(3)}(n, j, s) := E_{\tilde{g}_j}(s) - E_{\tilde{g}_{j-1}}(s),$$

for $s$ satisfying $s \in nI_{j-1}^{**} \cap nI_j^{**}$. The triangle inequality relates this quantity to the previous two and yields

$$\begin{aligned}
\left|\mathbb{D}_g^{(3)}(n, j, s)\right| &= \left|E_{\tilde{g}_j}(s) - E_{\tilde{g}_{j-1}}(s-1)\right| \\
&\leq \left|E_{\tilde{g}_j}(s) - E_{\tilde{g}_j}(s-1)\right| + \left|E_{\tilde{g}_j}(s-1) - E_{\tilde{g}_{j-1}}(s-1)\right| \\
&= \left|\mathbb{D}_g^{(1)}(n, j, s)\right| + \left|\mathbb{D}_g^{(2)}(n, j, s-1)\right|.
\end{aligned}$$

Hence to bound $S(\hat{g})$, we only need to derive upper bounds for $|\mathbb{D}_g^{(1)}|$ and $|\mathbb{D}_g^{(2)}|$, which we refer to as the *type-1* and *type-2 differences*, respectively. In Section 1.5.2 and 1.5.3, we show that both quantities are at most $S_g^*(2n)/n^{1-\lambda}$. Given this, and a Poisson-sampling McDiarmid's inequality derived in the next section, we establish the third inequality in Theorem 4.

### 1.5.1 From bounded difference to concentration

In this section, we establish a McDiarmid's inequality for Poisson sampling, showing that small sensitivity still implies strong concentration under formulation. We believe that this result is of independent interest. Specifically, we show that for any $p \in \Delta_k$, $N \sim \text{Poi}(n)$, $X^N \sim p$,

**Lemma 12.** *For any error parameter $\varepsilon \in (0, 1)$ and estimator $\hat{f}$ satisfying $S(\hat{f}) \geq 1/n$,*

$$\Pr\left(\left|\hat{f}(X^N) - \mathbb{E}\left[\hat{f}(X^N)\right]\right| > \varepsilon\right) \leq 4\exp\left(-\frac{\varepsilon^2}{2n(4S(\hat{f}))^2}\right).$$

*Proof.* By the linearity of expectation and triangle inequality,

$$|\mathbb{E}[\hat{f}(X^m)] - \mathbb{E}[\hat{f}(X^{m+1})]| \leq S(\hat{f}), \forall m.$$

Therefore for any $m$,

$$\begin{aligned}
|\mathbb{E}[\hat{f}(X^m)] - \mathbb{E}[\hat{f}(X^N)]| &= \left|\sum_{t=0}^{\infty} \mathbb{E}[\hat{f}(X^t)] \cdot \Pr(N = t) - \mathbb{E}[\hat{f}(X^m)]\right| \\
&= \left|\sum_{t=0}^{\infty} (\mathbb{E}[\hat{f}(X^t)] - \mathbb{E}[\hat{f}(X^m)]) \cdot \Pr(N = t)\right| \\
&\leq S(\hat{f}) \sum_{t=0}^{\infty} |t - m| \cdot \Pr(N = t).
\end{aligned}$$

We consider the last summation and simplify it as follows:

$$\sum_{t=0}^{\infty} |t - m| \cdot \Pr(N = t)$$

$$= \sum_{t=0}^{m} (m - t) \Pr(N = t) + \sum_{t=m}^{\infty} (t - m) \Pr(N = t)$$

$$= m \Pr(N \leq m) - \sum_{t=0}^{m} t \exp(-n) \frac{n^t}{t!} + \sum_{t=m}^{\infty} t \exp(-n) \frac{n^t}{t!} - m \Pr(N \geq m)$$

$$= m \Pr(N \leq m) - n \Pr(N \leq m - 1) + n \Pr(N \geq m - 1) - m \Pr(N \geq m)$$

$$= (m - n)(\Pr(N \leq m) - \Pr(N \geq m)) + n(\Pr(N = m) + \Pr(N = m - 1)).$$

Note that the second quantity on the right-hand side satisfies

$$\Pr(N = m) + \Pr(N = m - 1) \leq \Pr(N = n) + \Pr(N = n - 1)$$

$$\leq 2 \exp(-n) \frac{n^n}{n!}$$

$$\leq \frac{1}{\sqrt{n}}.$$

Consequently we have

$$|\mathbb{E}[\hat{f}(X^m)] - \mathbb{E}[\hat{f}(X^N)]| \leq S(\hat{f}) \sum_{t=0}^{\infty} |t - m| \cdot \Pr(N = t)$$

$$\leq S(\hat{f}) \left( (m - n)(\Pr(N \leq m) - \Pr(N \geq m)) + \sqrt{n} \right)$$

$$\leq S(\hat{f}) \cdot (|m - n| + \sqrt{n}).$$

Next, let $\varepsilon' \in (0, 1)$ be a constant to be determined later. The probability of interest satisfies

$$\Pr\left( \left| \hat{f}(X^N) - \mathbb{E}\left[ \hat{f}(X^N) \right] \right| > \varepsilon \right)$$

$$= \sum_{m=0}^{\infty} \Pr\left( \left| \hat{f}(X^m) - \mathbb{E}[\hat{f}(X^N)] \right| > \varepsilon \right) \Pr(N = m)$$

$$\leq \Pr(N \notin n[1 - \varepsilon', 1 + \varepsilon']) + \sum_{m \in n[1-\varepsilon', 1+\varepsilon']} \Pr\left( \left| \hat{f}(X^m) - \mathbb{E}[\hat{f}(X^N)] \right| > \varepsilon \right) \Pr(N = m).$$

We can easily bound the first term through the Chernoff bound. For the second term,

$$\sum_{m \in n[1-\varepsilon', 1+\varepsilon']} \Pr\left( \left| \hat{f}(X^m) - \mathbb{E}[\hat{f}(X^m)] \right| > \varepsilon - \left| \mathbb{E}[\hat{f}(X^m)] - \mathbb{E}[\hat{f}(X^N)] \right| \right) \Pr(N = m)$$

$$\leq \sum_{m \in n[1-\varepsilon', 1+\varepsilon']} \Pr\left( \left| \hat{f}^*(X^m) - \mathbb{E}[\hat{f}^*(X^m)] \right| > \varepsilon - S(\hat{f})(n\varepsilon' + \sqrt{n}) \right) \Pr(N = m)$$

$$\leq 2 \exp\left( -\frac{(\varepsilon - S(\hat{f})(n\varepsilon' + \sqrt{n}))^2}{n(1 + \varepsilon')(S(\hat{f}))^2} \right),$$

where the last step follows from the McDiarmid's inequality. Next, setting

$$\varepsilon' = \frac{\varepsilon}{2nS(\hat{f})} \in \left( 0, \frac{1}{2} \right),$$

we can rewrite last term, with the multiplicative factor of 2 removed, as

$$\exp\left( -\frac{(\varepsilon - S(\hat{f})(n\varepsilon' + \sqrt{n}))^2}{n(1 + \varepsilon')(S(\hat{f}))^2} \right) = \exp\left( -\frac{(\frac{\varepsilon}{2} - \sqrt{n}S(\hat{f}))^2}{n(1 + \varepsilon')(S(\hat{f}))^2} \right).$$

Hence, it suffices to obtain tight upper bounds on the right-hand side quantity, for which we consider the following two cases. If the parameter $\varepsilon$ is relatively large such that

$$\varepsilon \geq 4\sqrt{n}S(\hat{f}),$$

the quantity of interest is at most

$$\exp\left(-\frac{(\frac{\varepsilon}{2} - \sqrt{n}S(\hat{f}))^2}{n(1 + \varepsilon')(S(\hat{f}))^2}\right) \leq \exp\left(-\frac{\varepsilon^2}{32n(S(\hat{f}))^2}\right).$$

Otherwise, we have $\varepsilon^2/(32(S(\hat{f}))^2) \leq 1/2$, implying

$$2\exp\left(-\frac{\varepsilon^2}{32n(S(\hat{f}))^2}\right) \geq 2\exp\left(-\frac{1}{2}\right) > 1.$$

Consolidating previous results, we get

$$\Pr\left(\left|\hat{f}^*(X^{N''}) - \mathbb{E}\left[\hat{f}^*(X^{N''})\right]\right| > \varepsilon\right)$$

$$\leq 2\exp\left(-\frac{\varepsilon^2}{32n(S(\hat{f}))^2}\right) + \Pr(N'' \notin n[1 - \varepsilon', 1 + \varepsilon'])$$

$$\leq 2\exp\left(-\frac{\varepsilon^2}{32n(S(\hat{f}))^2}\right) + 2\exp\left(-\frac{1}{3}n\varepsilon'^2\right)$$

$$\leq 2\exp\left(-\frac{\varepsilon^2}{32n(S(\hat{f}))^2}\right) + 2\exp\left(-\frac{\varepsilon^2}{12n(S(\hat{f}))^2}\right)$$

$$\leq 4\exp\left(-\frac{\varepsilon^2}{32n(S(\hat{f}))^2}\right). \qquad \square$$

### 1.5.2 Bounding the type-1 difference

The following lemma provides tight upper bound on the type-1 difference.

**Lemma 13.** *For $s$ satisfying $s - 1, s,$ or $s + 1 \in nI_j^{**}$,*

$$\left|E_{\tilde{g}_j}(s) - E_{\tilde{g}_j}(s - 1)\right| \leq \frac{d_n \cdot 2^{4.5d_n + 1}}{n}L_g^*(2n).$$

*Proof.* Recall that

$$h_{v,x_j}(s) = \sum_{l=0}^{v}\binom{v}{l}(-x_j)^{v-l}\prod_{l'=0}^{l-1}\left(\frac{s}{n} - \frac{l'}{n}\right).$$

The difference between $h_{v,x_j}(s)$ and $h_{v,x_j}(s - 1)$ is

$$h_{v,x_j}(s) - h_{v,x_j}(s - 1) = \sum_{l=0}^{v}\binom{v}{l}(-x_j)^{v-l}\left(\prod_{l'=0}^{l-1}\left(\frac{s}{n} - \frac{l'}{n}\right) - \prod_{l'=0}^{l-1}\left(\frac{s-1}{n} - \frac{l'}{n}\right)\right)$$

$$= \sum_{l=0}^{v}\binom{v}{l}(-x_j)^{v-l}\left(\prod_{l'=0}^{l-1}\left(\frac{s}{n} - \frac{l'}{n}\right) - \prod_{l'=1}^{l}\left(\frac{s}{n} - \frac{l'}{n}\right)\right)$$

$$= \sum_{l=0}^{v}\binom{v}{l}(-x_j)^{v-l}\left(\frac{s}{n}\prod_{l'=1}^{l-1}\left(\frac{s}{n} - \frac{l'}{n}\right) - \frac{s-l}{n}\prod_{l'=1}^{l-1}\left(\frac{s}{n} - \frac{l'}{n}\right)\right)$$

$$= \sum_{l=0}^{v}\frac{l}{n}\binom{v}{l}(-x_j)^{v-l}\prod_{l'=1}^{l-1}\left(\frac{s}{n} - \frac{l'}{n}\right)$$

$$= \frac{v}{n}\sum_{l=0}^{v-1}\binom{v-1}{l}(-x_j)^{(v-1)-l}\prod_{l'=0}^{l-1}\left(\frac{s-1}{n} - \frac{l'}{n}\right)$$

$$= \frac{v}{n}h_{v-1,x_j}(s - 1).$$

By Lemma 4 and the definition of $L_g^*(2n)$,

$$|a_{jv}| \leq 2^{3.5d_n} \cdot 2 \cdot \sup_{z_1, z_2 \in I_j^{**}} |g(z_1) - g(z_2)| \leq 2^{3.5d_n+1} L_g^*(2n)|I_j^{**}|.$$

Therefore, the quantity of interest satisfies

$$
\begin{aligned}
\left| E_{\tilde{g}_j}(s) - E_{\tilde{g}_j}(s-1) \right| &= \left| \sum_{v=0}^{d_n} a_{jv} |I_j^{**}|^{-v} \left( h_{v,x_j}(s) - h_{v,x_j}(s-1) \right) \right| \\
&= \left| \sum_{v=0}^{d_n} a_{jv} |I_j^{**}|^{-v} \frac{v}{n} h_{v-1,x_j}(s-1) \right| \\
&\leq \frac{2^{3.5d_n+1}}{n} L_g^*(2n)|I_j^{**}| \sum_{v=1}^{d_n} v|I_j^{**}|^{-v} \left( 2|I_j^{**}| \right)^{v-1} \\
&\leq \frac{2^{3.5d_n+1}}{n} L_g^*(2n) \sum_{v=1}^{d_n} v2^{v-1} \\
&\leq \frac{d_n \cdot 2^{4.5d_n+1}}{n} L_g^*(2n),
\end{aligned}
$$

where the third last inequality follows from Lemma 8 by setting $\delta = |I_j^{**}|$, and the last inequality follows from $\sum_{v=1}^{d_n} v2^{v-1} \leq d_n \cdot 2^{d_n}$. $\qquad \square$

### 1.5.3 Bounding the type-2 difference

In this section, we show the following upper bound on the type-2 difference.

**Lemma 14.** *For $s$ satisfying $s \in nI_{j-1}^{**} \cap nI_j^{**}$,*

$$\left| E_{\tilde{g}_{j-1}}(s) - E_{\tilde{g}_j}(s) \right| \leq \frac{4 \cdot 2^{4.5d_n}}{n} D_g^*(2n).$$

*Proof.* Note that $E_{\tilde{g}_{j-1}}(N_i) - E_{\tilde{g}_j}(N_i)$ is an unbiased estimator of $(\tilde{g}_{j-1} - \tilde{g}_j)(x)$. For simplicity, denote $\tilde{q}_j(x) := (\tilde{g}_{j-1} - \tilde{g}_j)(x)$ and $I_j^\Lambda := I_{j-1}^{**} \cap I_j^{**} = c_n \left[ (j-3)^2 \mathbb{1}_{j \geq 3}, (j+1)^2 \right]$. Then we have $|\tilde{q}_j(x)| \leq 2D_g^*(2n)/n$ for $x \in I_j^\Lambda$. Let

$$x_j' := c_n(j-3)^2 \mathbb{1}_{j \geq 3}$$

be the left end point of $I_j^\Lambda$, and

$$|I_j^\Lambda| := c_n(j+1)^2 - c_n(j-3)^2 \mathbb{1}_{j \geq 3}$$

be the length of $I_j^\Lambda$. For any $x \in I_j^\Lambda$, there exists $y_x \in [0, 1]$ such that

$$x = x_j' + |I_j^\Lambda| y_x.$$

Since $x \to y_x$ is a linear transformation, there exist coefficients $b_{jv}$, $v = 0, \ldots, d_n$, independent of $x$, such that

$$\tilde{q}_j(x) = \sum_{v=0}^{d_n} b_{jv} y_x^v.$$

By the definition of $\tilde{q}_j(x)$ and the triangle inequality, we can deduce that $|\tilde{q}_j(x)| \leq 2D_g^*(2n)/n$ for all $x \in I_j^\Lambda$. Furthermore, according to Lemma 4,

$$|b_{jv}| \leq \frac{2^{4.5d_n}}{n} D_g^*(2n).$$

Substituting $y_x$ by $|I_j^\Lambda|^{-1}(x - x_j')$, we can re-write $\tilde{q}_j(x)$ as

$$\tilde{q}_j(x) = \sum_{v=0}^{d_n} b_{jv} |I_j^\Lambda|^{-v} \left( x - x_j' \right)^v.$$

Consequently, we have the following equality:
$$\left(E_{\tilde{g}_{j-1}} - E_{\tilde{g}_j}\right)(s) = \sum_{v=0}^{d_n} b_{jv}|I_j^\Lambda|^{-v} h_{v,x_j'}(s).$$

Therefore, for all $s \in nI_j^\Lambda$,
$$
\begin{aligned}
\left|\left(E_{\tilde{g}_{j-1}} - E_{\tilde{g}_j}\right)(s)\right| &= \left|\sum_{v=0}^{d_n} b_{jv}|I_j^\Lambda|^{-v} h_{v,x_j'}(s)\right| \\
&\leq \sum_{v=0}^{d_n} \frac{2 \cdot 2^{3.5 d_n} D_g^*(2n)}{n} |I_j^\Lambda|^{-v} \left(2|I_j^\Lambda|\right)^v \\
&\leq \frac{2^{3.5 d_n} D_g^*(2n)}{n} \sum_{v=0}^{d_n} 2^{v+1} \\
&\leq \frac{4 \cdot 2^{4.5 d_n}}{n} D_g^*(2n). \qquad \square
\end{aligned}
$$

## 2 Proofs of other theorems

### 2.1 Proof of Theorem 5

Let $\vec{p} \in \Delta_k$ be an arbitrary distribution and $X^N$ be an i.i.d. sample sequence from $\vec{p}$ of an independent $N \sim \text{Poi}(2n)$ size. Applying sample splitting to $X^N$, we denote by $N_i$ and $N_i'$ the number of times symbol $i \in [k]$ appearing in the first and second sub-sample sequences, respectively. Applying the technique presented in Section 1.2, we can estimate the additive property
$$f(\vec{p}) = \sum_{i \in [k]} f_i(p_i)$$
by the estimator
$$\hat{f}^*(X^N) := \sum_{i \in [k]} \hat{f}_i^*(N_i, N_i').$$

We start by bounding the bias of $\hat{f}^*$. Fix $\lambda \in (0, 1/4)$ and let $T$ be a sufficiently large constant satisfying $T_1 \gg \max_{i \in [k]} \max_{x \in [0,1]} |f_i(x)|$. The results in Section 1.3 and triangle inequality imply

$$
\begin{aligned}
\left|\mathbb{E}[\hat{f}^*(X^N)] - f(\vec{p})\right| &= \left|\mathbb{E}\left[\sum_{i \in [k]} \hat{f}_i^*(N_i, N_i')\right] - \sum_{i \in [k]} f_i(p_i)\right| \leq \sum_{i \in [k]} \left|\mathbb{E}[\hat{f}_i^*(N_i, N_i')] - f_i(p_i)\right| \\
&\leq \sum_{i \in [k]} \left(\frac{T}{n^5} \cdot p_i + \frac{3T d_n \cdot 2^{4.5 d_n}}{c_n n^5} \cdot p_i + \frac{5}{n} D_{f_i}^*(2n, p_i)\right) \\
&= \frac{T}{n^5} + \frac{3T d_n \cdot 2^{4.5 d_n}}{c_n n^5} + \frac{5}{n} \sum_{i \in [k]} D_{f_i}^*(2n, p_i) \\
&\leq \frac{T}{n^5} + \frac{T n^\lambda}{c n^4 \log n} + \frac{5}{n} \sum_{i \in [k]} D_{f_i}^*(2n, p_i).
\end{aligned}
$$

Next we analyze the variance of $\hat{f}^*$. Due to Poisson sampling and sample splitting, all the counts $N_i$ and $N_i'$, $i \in [k]$ are mutually independent. Therefore, by Lemma 11 in Section 1.4,

$$
\begin{aligned}
\text{Var}(\hat{f}^*(X^N)) &= \text{Var}\left(\sum_{i \in [k]} \hat{f}_i^*(N_i, N_i')\right) = \sum_{i \in [k]} \text{Var}\left(\hat{f}_i^*(N_i, N_i')\right) \\
&\leq \sum_{i \in [k]} \left(\frac{72 c (\log n)}{n^{1-3\lambda}} \left(L_{f_i}^*(2n, p_i)\right)^2 \cdot p_i + \frac{8T^2}{n^5} \cdot p_i\right) \\
&= \frac{8T^2}{n^5} + \frac{72 c (\log n)}{n^{1-3\lambda}} \sum_{i \in [k]} \left(L_{f_i}^*(2n, p_i)\right)^2 \cdot p_i.
\end{aligned}
$$

To characterize higher-order central moments of $\hat{f}^*$, note that changing one sample point in $X^N$ would change the counts $N_i$, $N_i'$, or both for at most two symbols. Hence, according to Section 1.5, for a given $n$ the sensitivity of $\hat{f}^*$, also defined in the same section, satisfies

$$S(\hat{f}^*) \leq \frac{4 \max_{i \in [k]} S^*_{f_i}(2n)}{n^{1-\lambda}}.$$

This bound together with Lemma 12 yields

$$\Pr\left(\left|\hat{f}^*(X^N) - \mathbb{E}\left[\hat{f}^*(X^N)\right]\right| > \varepsilon\right) \leq 4 \exp\left(-\frac{n^{1-2\lambda}\varepsilon^2}{(32 \max_{i \in [k]} S^*_{f_i}(2n))^2}\right).$$

## 2.2 Proof of Theorem 1

Recall that an additive property $f$ is a Lipschitz property if all the $f_i$'s have uniformly bounded Lipschitz constants. Our proof of Theorem 1 relies on the following lemma, which corresponds to Theorem 7.2 in [3] whose proof is completely constructive. In other words, there is an explicit procedure to compute the polynomial described in the following lemma.

**Lemma 15.** *There exists a universal constant $C$ such that for any degree parameter $d \in \mathbb{Z}$ and 1-Lipschitz function $g$ over an arbitrary bounded interval $I := [x_1, x_2]$, one can find a polynomial $\tilde{g}$ of degree at most $d$ satisfying*

$$|\tilde{g}(x) - g(x)| \leq \frac{C\sqrt{|I|(x - x_1)}}{d}, \forall x \in I.$$

We restate Theorem 1 below under Poisson sampling. By the results in [9], this suffices to imply the corresponding result under fixed sampling, where the sample size is fixed to be $n$.

**Theorem 1.** *If $f$ is an $L$-Lipschitz property, then for any $\vec{p} \in \Delta_k$, $N \sim \text{Poi}(n)$, and $X^N \sim \vec{p}$,*

$$\left|\mathbb{E}\left[\hat{f}^*(X^N)\right] - f(\vec{p})\right| \lesssim \sum_{i \in [k]} L\sqrt{\frac{p_i}{n \log n}} \leq L\sqrt{\frac{k}{n \log n}},$$

*and*

$$\text{Var}(\hat{f}^*(X^N)) \leq \frac{L^2}{n^{1-4\lambda}}.$$

*Proof.* Without loss of generality, we assume that all the $f_i$'s have Lipschitz constants uniformly bounded by 1. The derivations in Section 1.3 and 2.1 imply

$$\left|\mathbb{E}[\hat{f}^*(X^N)] - f(\vec{p})\right| \lesssim \frac{1}{n^3} + \sum_{i \in [k]} \max_{j' \in [j_{p_i}-1, j_{p_i}+1]} |\tilde{f}_{i,j'}(p_i) - f(p_i)|,$$

Here, for $j' > 3$, we choose $\tilde{f}_{i,j'}(x)$ to be the min-max polynomial defined in Section 1.2; for $j' \leq 3$, we employ the polynomials used in Lemma 15 instead. Note that the latter polynomials may not be the min-max polynomials. However, this would not affect our analysis as our proof in Section 1 also holds for these polynomials (simply change the definition of $D^*_g(2n, p)$).

For any symbol $i$ satisfying $j_{p_i} \leq 3$,

$$\max_{j' \in [j_{p_i}-1, j_{p_i}+1]} |\tilde{f}_{i,j'}(p_i) - f_i(p_i)| \lesssim \max_{j' \in [j_{p_i}-1, j_{p_i}+1]} \frac{\sqrt{|I_{j'}|(p_i - 0)}}{d_n} \asymp \frac{\sqrt{\frac{\log n}{n}p_i}}{\log n} = \sqrt{\frac{p_i}{n \log n}}.$$

On the other hand, applying Lemma 15 and the definition of min-max polynomials to our case implies that for any symbol $i$ satisfying $j_{p_i} > 3$,

$$\max_{j' \in [j_{p_i}-1, j_{p_i}+1]} |\tilde{f}_{i,j'}(p_i) - f_i(p_i)| \lesssim \max_{j' \in [j_{p_i}-1, j_{p_i}+1]} \frac{|I_{j'}|}{d_n} \asymp \frac{j_{p_i}}{n}$$

and

$$p_i \in I^{**}_{p_i} = c\frac{\log n}{n}[(j_{p_i} - 3)^2, (j_{p_i} + 2)^2],$$

or equivalently,

$$j_{p_i} \in \left[ \sqrt{\frac{np_i}{c \log n}} - 2, \sqrt{\frac{np_i}{c \log n}} + 3 \right] \subseteq \left[ \sqrt{\frac{np_i}{c \log n}} - 2, 4\sqrt{\frac{np_i}{c \log n}} \right].$$

Therefore,

$$\max_{j' \in [j_{p_i}-1, j_{p_i}+1]} |\tilde{f}_{i,j'}(p_i) - f_i(p_i)| \lesssim \frac{j_{p_i}}{n} \leq \frac{1}{n} \cdot 4\sqrt{\frac{np_i}{c \log n}} \lesssim \sqrt{\frac{p_i}{n \log n}}.$$

The above result together with the Cauchy-Schwarz inequality implies

$$\left| \mathbb{E}[\hat{f}^*(X^N)] - f(\vec{p}) \right| \lesssim \frac{1}{n^3} + \sum_{i \in [k]} \sqrt{\frac{p_i}{n \log n}} \leq 2 \sum_{i \in [k]} \sqrt{\frac{p_i}{n \log n}} \leq 2\sqrt{\frac{k}{n \log n}},$$

where the second inequality follows by observing $\sqrt{a+b} \leq \sqrt{a} + \sqrt{b}$. Analogously, by previous results, we can bound the variance of $\hat{f}^*$ as follows,

$$\mathrm{Var}(\hat{f}^*(X^N)) \lesssim \frac{1}{n^5} + \frac{\log n}{n^{1-3\lambda}} \sum_{i \in [k]} \left( L_{f_i}^*(2n, p_i) \right)^2 \cdot p_i.$$

By the definition of $L_{f_i}^*$ and the assumption that $f_i$ is 1-Lispchitz,

$$L_{f_i}^*(2n, p_i) = \max_{j' \in [j_{p_i}-1, j_{p_i}+1]} \left\{ \sup_{x,y \in I_{j'}^{**}, |y-x| \geq 1/n} \frac{|f_i(y) - f_i(x)|}{|y - x|} \right\} \leq 1.$$

Hence,

$$\mathrm{Var}(\hat{f}^*(X^N)) \lesssim \frac{1}{n^5} + \frac{\log n}{n^{1-3\lambda}} \sum_{i \in [k]} p_i \leq \frac{1}{n^5} + \frac{\log n}{n^{1-3\lambda}} \leq \frac{1}{n^{1-4\lambda}}. \qquad \square$$

## 2.3  Private property estimation

According to [1], we can construct a differentially private property estimator $\hat{f}_{DP}^*$ by first applying $\hat{f}^*$ to the sample sequence, and then privatizing its estimate through adding Laplace noise. The following lemma characterizes the sample complexity of $\hat{f}_{DP}^*$, and enables us to upper bound the private sample complexity of estimating general additive properties.

**Lemma 2.** *There exists a universal constant $c^*$ such that*

$$C_f(\hat{f}_{DP}^*, 2\varepsilon, 1/3, 2\alpha) \leq \frac{c^*}{4} \left\{ C_f(\hat{f}^*, \varepsilon, 1/3) + \min \left\{ n : \ S(\hat{f}^*, n) \leq \varepsilon\alpha \right\} \right\}.$$

*The right-hand side also upperly bounds $C_f(2\varepsilon, 1/3, 2\alpha)$.*

By Theorem 5 in the main paper, for $\hat{f}^*(X^n)$ to achieve an accuracy of $\varepsilon$ with probability at least $2/3$, for all $\vec{p} \in \Delta_k$, it suffices for the sampling parameter $n$ to satisfy the following three conditions:

$$n \geq \left( \frac{4T}{\varepsilon} \right)^{\frac{1}{3}}, \quad \frac{n}{D_{f_i}^*(n)} \geq \frac{20k}{\varepsilon}, \quad \text{and} \quad \frac{n^{\frac{1}{2}-\lambda}}{S_{f_i}^*(n)} \geq \frac{16\sqrt{\log 12}}{\varepsilon}, \forall i.$$

where $T$ is a uniform upper bound on $|f_i(x)|, \forall i \in [k], x \in [0,1]$. To further make the estimator's $n$-sensitivity smaller than $\alpha\varepsilon$, the sampling parameter $n$ should also satisfy the condition:

$$\frac{n^{1-\lambda}}{S_{f_i}^*(n)} \geq \frac{1}{\alpha\varepsilon}, \forall i.$$

Define $n_f(2\varepsilon, 2\alpha)$ as the smallest $n$ satisfying all the four inequalities above. Then,

**Theorem 6.** *The $(\varepsilon, 1/3, \alpha)$-private sample complexity for any additive property $f$ satisfies*

$$C_f(\varepsilon, 1/3, \alpha) \leq n_f(\varepsilon, \alpha).$$

## 2.4 High-probability property estimation

In this section, we present tight upper and lower bounds on the $(\varepsilon, \delta)$-sample complexity of estimating various properties. The error parameter $\varepsilon$ can take any value in $(0, 1)$. All the upper bounds follow from Theorem 5 in the main paper. Below we focus on deriving the lower bounds.

### Shannon entropy

For any absolute constant $\beta \in (0, 1)$,

$$\frac{k}{\varepsilon \log k} + \log \frac{1}{\delta} \cdot \frac{\log^2 k}{\varepsilon^2} \lesssim C_f(\varepsilon, \delta).$$

The first part of the lower bound follows directly from [9]. To show the second part of the lower bound, let $\varepsilon' \in (0, 1)$ be a parameter to be determined later, and consider the following [9] two distributions in $\Delta_k$,

$$\vec{p_1} := \left( \frac{1 - \varepsilon'}{3(k-1)}, \dots, \frac{1 - \varepsilon'}{3(k-1)}, \frac{2 + \varepsilon'}{3} \right)$$

and

$$\vec{p_2} := \left( \frac{1}{3(k-1)}, \dots, \frac{1}{3(k-1)}, \frac{2}{3} \right).$$

The entropy difference between these two distributions is

$$\begin{aligned}
H(\vec{p_2}) - H(\vec{p_1}) &= \frac{1 - \varepsilon'}{3} \log \frac{1 - \varepsilon'}{3(k-1)} + \frac{2 + \varepsilon'}{3} \log \frac{2 + \varepsilon'}{3} - \frac{1}{3} \log \frac{1}{3(k-1)} - \frac{2}{3} \log \frac{2}{3} \\
&= \frac{\varepsilon'}{3} \log(2(k-1)) + \frac{1 - \varepsilon'}{3} \log(1 - \varepsilon') + \frac{2 + \varepsilon'}{3} \log \frac{2 + \varepsilon'}{2} \\
&\geq \frac{\varepsilon'}{3} \log(2e^{-1}(k-1)).
\end{aligned}$$

For sufficiently large $k$, choose $\varepsilon' = 9\varepsilon / \log(2e^{-1}(k-1))$. The difference between $H(\vec{p_1})$ and $H(\vec{p_2})$ is at least $3\varepsilon$.

On one hand, since $\vec{p_1}$ and $\vec{p_2}$ differ by $2\varepsilon'/3$ in $\ell_1$ distance, any algorithm that distinguishes the two distributions with confidence $1 - \delta$ requires at least $\Omega(\frac{1}{\varepsilon'^2} \log \frac{1}{\delta})$ samples. On the other hand, any entropy estimator $\hat{f}$ that utilizes $C_f(\hat{f}, \varepsilon, \delta)$ samples can be used to distinguish $\vec{p_1}$ and $\vec{p_2}$ with confidence $1 - \delta$. This yields the desired lower bound.

### Normalized support size

The lower bound follows from [8].

### Power sum

For any absolute constants $\beta \in (0, 1)$ and $a \in (1/2, 1)$,

$$\frac{k^{\frac{1}{a}}}{\varepsilon^{\frac{1}{a}} \log k} + \log \frac{1}{\delta} \cdot \frac{k^{2-2a}}{\varepsilon^2} \lesssim C_f(\varepsilon, \delta)$$

and

$$C_f(\varepsilon, \delta) \lesssim \frac{k^{\frac{1}{a}}}{\varepsilon^{\frac{1}{a}} \log k} + \left[ \left( \log \frac{1}{\delta} \cdot \frac{1}{\varepsilon^2} \right)^{\frac{1}{2a-1}} \right]^{1+\beta}.$$

The first part of the lower bound follows from [5]. Analogously, to show the second part of the lower bound, let $\varepsilon'' \in (0, 1)$ be a parameter to be determined later, and consider the following two distributions in $\Delta_k$,

$$\vec{p_3} := \left( \frac{1 - \varepsilon''}{3(k-1)}, \dots, \frac{1 - \varepsilon''}{3(k-1)}, \frac{2 + \varepsilon''}{3} \right)$$

and

$$\vec{p_2} := \left( \frac{1}{3(k-1)}, \dots, \frac{1}{3(k-1)}, \frac{2}{3} \right).$$

The difference between the power sums of these two distributions satisfies

$$P_a(\vec{p_2}) - P_a(\vec{p_3}) = (k-1)\left(\frac{1}{3(k-1)}\right)^a + \left(\frac{2}{3}\right)^a - (k-1)\left(\frac{1-\varepsilon''}{3(k-1)}\right)^a - \left(\frac{2+\varepsilon''}{3}\right)^a$$

$$= \frac{(k-1)^{1-a}}{3^a}\left(1 - (1-\varepsilon'')^a\right) + \left(\frac{2}{3}\right)^a - \left(\frac{2+\varepsilon''}{3}\right)^a$$

$$\geq \frac{(k-1)^{1-a}}{3^a}a\left(\varepsilon'' - \frac{\varepsilon''^2}{2}\right) + \left(\frac{2}{3}\right)^a\left(1 - a\left(1 + \frac{\varepsilon''}{2}\right)\right)$$

$$\geq \frac{a\varepsilon''}{2\cdot 3^a}\left((k-1)^{1-a} - 2^a\right).$$

For $k$ that is sufficiently large, choose parameter $\varepsilon'' = 6\varepsilon \cdot 3^a / \left(a(k-1)^{1-a} - a\cdot 2^a\right)$. The difference between $P_a(\vec{p_2})$ and $P_a(\vec{p_3})$ is at least $3\varepsilon$.

The desired lower bound follows from the same reasoning as in the Shannon-entropy case.