[Reviews · NeurIPS 2019]

Reviewer 1



Originality: While using polynomial approximators for functionals of distributions has a long history, the "local" sense in which this paper proposes using them is interesting. The discussion of prior work in the context of the paper's results could use more elaboration; see below. Quality: I have not checked all the proofs, but the ones I did check seem correct. Clarity: The paper is reasonably well-written and intuition is presented well. Significance: While the methodology is interesting and the applications to various estimation problems seems promising, the paper would benefit from a revision that better articulates certain specific conclusions. Details below: -- The paper left me slightly confused about the correct asymptotic to consider in order to compare the various methodologies for these problems. In particular, the results of the current paper seem to exhibit a variance term that has a strictly sub-optimal exponent of the sample size n. In the case where k is fixed and n \to \infty, this should imply that the estimators in the current paper are not actually minimax optimal? Optimality is of course evident in the case where k grows, but this is not mentioned anywhere in the manuscript. It would be good to be up front about this. -- In the absence of concrete experiments for some of these properties, it is not clear that this methodology is actually more suited to the various properties than existing estimators, so an empirical evaluation (even just on synthetic data) would be very useful. In parituclar, many of the claimed imporvements over prior art are (sub-)logarithmic, and so it is not immediately clear that employing this methodology provides practical gains in useful regimes of n. Specific comments: -- It would be good to make the Lipschitz constant explicit in the statement of Theorem 1. -- In table 2, I think it would be helpful from a comparison perspective if the best available bounds for these three properties (before this paper) were also included, so as to facilitate a concrete comparison. -- Line 101: plug -> plugging -- Line 234: liner -> linear -- Line 277: Number of symbols is at most k, correct? The draft says n. In general, this subsection is slightly confusing and should be rewritten for clarity.

Reviewer 2



This paper considers property estimation for discrete distributions: given a "smooth enough" functional f of distributions, assumed further to be "additive" (i.e., f(p) = sum_i f_i(p_i), where p=(p_i)_i is the distribution identified to its pmf), the authors derive a general approach, based on polynomial approximation, to obtain optimal or near-optimal rates, with estimators that are easy to construct and can be computed in time nearly linear in the sample size. The main result is Theorem 1, which provides the general upper bound statement. Overall, I find the results and approach of this paper nice, but somewhat incremental; and far from the "novelty" and "strong implications" claimed. (Also, the privacy part doesn't really seem to fit with the rest... basically, you get Lipscitz estimators, so can get privacy almost for free; maybe make that a distinct section, instead of putting it in the middle, as I'd argue it's not really the meat of your work?) Comments: - line 73: distance estimation to a fixed distribution or between two unknown distributions is known as well by the techniques of Valiant-Valiant; see https://theory.stanford.edu/~valiant/papers/VV_LinEst.pdf ***- ll.93-96: This is one of my main criticisms. You claim a novel approach, departing conceptually from the two extisting methods you described before. However, what you described here does not seem really novel: it is, unless I misunderstood what you describe, a refinement of the "approximation" technique, so sounds a lot more incremental than what you suggest. - ll 127-133 on high probability: this is really misleading. As far as I can tell, the lack of high-probability statements from previous work is not that the approaches do not yield the "good" (additive) dependence on log(1/\delta): for all we know, they might (using McDiarmid, or even direct arguments?), or might not -- it's just that it wasn't *considered* in those works. Thus, claiming that your work significantly improves on those wrt delta by comparing your results with the previous one *using the median trick* is not something of a misrepresentation. - ll. 140-141: so you need subconstant approximation parameter eps for support size? That's a *big* restriction. - l.150: how "conventional" is that? Is it actually common (I don't recall seeing this assumption). Also, do you assume knowledge of that constant c by the algorithm? - l.154-155: you should compare with the upper bound of [2]! They have a result on this, and your upper bounds are nearly the same (you improve by basically a (log k)^2 factor in the middle term). You cannot not mention it! (also, please remove the constants 2 in 2eps, 2alpha, they are not meaningful and obscure the statement, since you use \lesssim afterwards) - l.234: "liner" -> "linear" UPDATE: I have read the authors' response, and taken it into account in my assessment.

Reviewer 3



The paper considers the problem of distribution property estimation, where you observe n i.i.d samples from a distribution discrete distribution p over [k] = {1, 2, ...,k} and you want to estimate a certain property f(p) of the distribution. The problem is a fundamental problem in statistics and machine learning with an increasing interest in the finite sample regime where you consider the sample complexity during the recent decade. The paper considers all distributions that can be expressed in an additive form with f(p) = \sum_i f_i(p_i) and provides a unified algorithm by approximating each f_i by a piece-wise polynomial function f'_i and the final estimation is the sum of unbiased estimators for f'_i(p_i)'s. The proposed estimator: 1) achieves a sublinear sample complexity of (k/\log k) for all Lipschitz properties in near-linear time. 2) achieve optimal sample complexity for various symmetric properties including Shannon entropy, support size, support cover, distance to a certain distribution and etc. Although sample optimal estimators for the aforementioned properties already exist in the literature. But it is not clear whether they can be extended to other all additive properties. The idea of using best polynomial approximation to estimate additive properties is not new (See [Wu, Yang, 2016] for entropy estimation). This paper extends the idea to a more general class of distribution properties by applying best polynomial approximation in multiple intervals. And this extension might be of independent interest. Cons: 1. The writing of the paper needs to be improved. It might be good to write what the final estimator will be in Section 2 to give an overall view of the whole algorithm before going into math details in the following sections. 2. Detailed comments´╝Ü 1) The abbreviation PML is used without defining. 2) Line 81: belongs to 3) The bounds for power sum seem not matching. Can you elaborate more on this? 4) The equation between 154 and 155 is missing a term \frac{\log^2{\min{n,k}}}{\alpha^2}. 5) Line 214. The statement may not be correct without proper assumption on the posterior of p although I believe lien 213 is correct. 6) Line 237. Should be bias instead of error. 7) The proof for theorem 3 is missing in the appendix. Overall, I would recommend the paper for acceptance.

[Author Response · NeurIPS 2019]

We thank the reviewers for their valuable and helpful comments. Below we address the comments sequentially.

Reviewer #1: *Correct asymptotic* For the properties considered in the manuscript, even the naive empirical-frequency estimator is sample-optimal in the *large-sample* regime (termed "simple regime" in [25]) where the number of samples $n$ far exceeds the alphabet size $k$. The interesting regime, addressed in numerous recent publications [12, 14, 22, 24, 26], is where $n$ and $k$ differ by at most a logarithmic factor. In this range, $n$ is sufficiently small that sophisticated techniques can help, yet not too small that nothing can be estimated. Since $n$ and $k$ are given, one can decide whether the naive estimator suffices, or sophisticated estimators are needed. Thank you for suggesting and we will make this clear.

*Absence of concrete experiments* Thank you for asking about the practicality of logarithmic improvements. While some complexity domains look for exponential improvements, for data collection even constant reduction in the number of samples is significant to practitioners. This is one of several recent papers that show a logarithmic reduction over standard estimators. The manuscript does not show new experiments as its main contribution is to theoretically solidify this improvement by showing that it can be achieved by a unified estimator. Some of the results we get are significant. For example, Theorem 1 on Lipschitz property estimation is the most general result we know on this topic.

Specific comments: *Explicit Lipschitz constant in Theorem 1* Sure, the constant is clear from the proof, we'll add it.

*Comparisons in Table 2* All the results are ready and we will add comparisons. Line 101 and 234: typos corrected.

Line 277: *Number of symbols is at most $k$, correct?* Correct, it is also at most $n$ here since we consider symbols in $X^n$. We will rewrite this paragraph to improve its clarity. Thank you for suggesting.

Reviewer #2: *Results are nice but somewhat incremental* While polynomial approximation has been applied to property estimation, the use of piece-wise polynomials is novel and the analysis of the resulting algorithm is nontrivial (e.g., supplement's Section 2.3). Some of the results are also novel and significant. For example, line 120 shows that all Lipschitz properties can be estimated up to an error of $\varepsilon$ using $k/(\varepsilon^2 \log k)$ samples, regardless of symmetry.

*The privacy part doesn't fit* and *compare with the upper bound of [2] in line 154-155* Please view Theorem 5 as the main result and all the other results as its corollaries. For privacy, our contribution is a unified approach, not the known sample complexity bounds. We mentioned, in line 65-66, that [2] derived tight lower and upper bounds. We apologize for making a mistake and not mentioning the upper bounds in [2] again in line 154-155. We will mention them again in line 154 and clarify that these bounds are known. In addition, we have removed constants 2 in $2\varepsilon$ and $2\alpha$.

Comments: Line 73: *Distance estimation using the Valiant-Valiant techniques* Sure, we will introduce this result.

Line 93-96: *The approach is not conceptually different* We provide a different view of property estimation that allows us to simplify the proofs and broaden the range of the results. We will extend the paragraph and make our point clearer.

Line 127-133: *Previous works may also lead to high-probability statements* The major contribution here is a unified approach to deriving high-probability estimators. We are not claiming other methods can not achieve these guarantees, instead, we want to demonstrate that our method has many desired attributes. We compared our result with the median trick approach since it is a natural baseline. We will check related works and clarify this point.

Line 140-141: *Subconstant $\varepsilon$ for support size* We need this condition only to derive the simple upper bound presented in Table 2. This is not required by our algorithm, which achieves the minimax MSE for support size estimation.

Line 150: *Conditions for KL divergence estimation* These conditions appear in "Minimax rate-optimal estimation of KL divergence between discrete distributions" and some subsequent papers. We will include these references.

Reviewer #3: *Present the estimator in Section 2* Sure, we will include a high-level description of the final algorithm.

*The abbreviation PML is not defined* We use "PML" to refer to a different (also well-known) method. The definition of this "PML estimator" is actually quite simple and intuitive, and we will include it. Line 81: belongs –> belongs to.

*The bounds for power sum not matching* The lower and upper bounds in Table 2 match with each other whenever the first term dominates. We believe that the upper bound is not tight and a finer analysis of our estimator may tighten it.

Line 154-155: *Missing a term* We have added the missing term to the expression.

Line 214: *Statement may be incorrect* This statement should be correct. Basically, we view probability as "degrees of belief", and if we always claim $p \in I_j^*$ whenever $\hat{p}_1 \in I_j$, then we will be correct with high probability. Note that this explanation only provides intuition and is not used in the proof. We will think about how to further clarify.

Line 237: *Should be 'bias' instead of 'error'* Yes, we have corrected this. Thank you for suggesting.

*The proof for Theorem 3 is missing in the appendix* The proof of Theorem 3 is a direct application of Theorem 5. For each property, we need only a few lines to establish the result. We will provide this short proof in the supplement.

[Meta-Review · NeurIPS 2019]

The paper consider estimating additive, Lipschitz properties of distributions and proposes a new technique based on a piecewise polynomial approximation. The approach is fairly general (in particular it is applicable to asymmetric properties, while PML-type approaches won't be a good fit) and gives near-optimal results in the regime of interest k, n \to \infty. We had a long discussion about this paper. On the positive side: 1. The results are interesting, especially that we get estimators for asymmetric properties and with high-probability guarantees. 2. The techniques are definitely non-trivial, but seem somewhat natural/incremental in light of global polynomial approximation ideas already in the literature. But given that the proofs are non-trivial, it seems useful to have an analysis for local polynomial approximation estimators worked out, and the paper does this with generality. On the negative side: 1. The conceptual main message of the paper seems confused. My sense is that local polynomial approximations is the main message, but the paper spends substantial time on near-linear time, privacy and several other distractions. This makes it much harder to figure out what's going on. 2. There is some overselling/over-claiming going on, as the reviewers have pointed out. Specifically sub-optimal dependencies in the "lower order" terms are largely ignored (the rebuttal provides a compelling answer to this, and I suggest adding it to the paper) and some important prior results are omitted from the discussion. Despite these negatives, the reviewers and I believe the pros outweigh the cons here, so we are recommending acceptance. I strongly encourage the reviewers to do a thorough re-writing of the paper for the camera ready, focusing on the a simple conceptual message.